# Molecular networking and computational NMR analyses uncover six polyketide-terpene hybrids from termite-associated *Xylaria* isolates
Seoung Rak Lee[1,2,9], Marie Dayras [3,9], Janis Fricke [4], Huijuan Guo[4], Sven Balluff[3], Felix Schalk[4], Jae Sik Yu[1,5], Se Yun Jeong[1], Bernd Morgenstern[6], Bernard Slippers[7], Christine Beemelmanns [3,4,8,10] ✉ & Ki Hyun Kim [1,10] ✉

Fungi constitute the Earth's second most diverse kingdom, however only a small percentage of these have been thoroughly examined and categorized for their secondary metabolites, which still limits our understanding of the ecological chemical and pharmacological potential of fungi. In this study, we explored members of the co-evolved termite-associated fungal genus *Xylaria* and identified a family of highly oxygenated polyketide-terpene hybrid natural products using an MS/MS molecular networking-based dereplication approach. Overall, we isolated six no yet reported xylasporin derivatives, of which xylasporin A (**1**) features a rare cyclic-carbonate moiety. Extensive comparative spectrometric (HRMS[2]) and spectroscopic (1D and 2D NMR) studies allowed to determine the relative configuration across the xylasporin family, which was supported by chemical shift calculations of more than 50 stereoisomers and DP4+ probability analyses. The absolute configuration of xylasporin A (**1**) was also proposed based on TDDFT-ECD calculations. Additionally, we were able to revise the relative and absolute configurations of co-secreted xylacremolide B produced by single x-ray crystallography. Comparative genomic and transcriptomic analysis allowed us to deduce the putative biosynthetic assembly line of xylasporins in the producer strain X802, and could guide future engineering efforts of the biosynthetic pathway.

The fungal kingdom ranks as the second most diverse kingdom on Earth, estimated to comprise 3–4 million species[1]. Within this kingdom, Sordariomycetes stands out as a diverse and significant class, including fungal families such as Hypoxylaceae and Xylariaceae[1,2] which have recently gained considerable attention due to their enormous biosynthetic repertoire to produce secondary metabolites with diverse biological activities[3,4]. While the Xylariaceae family includes predominately free-living saprophytic members that are often noticeable due to the development of stromatal tissue on decaying wood, it also includes a distinctive *Xylaria* subgenus, previously termed *Pseudoxylaria*, whose member have been isolated from fungal garden material of fungus-farming termites, in which termites cultivate the fungal mutualist *Termitomyces* as a crop[5–7], Members of the co-evolved *Xylaria* subgenus exclusively appear in the fungus comb of termite colonies that are weakened or abandoned, rapidly dominating any remaining termite food source (Fig. 1)[8,9],

[1]School of Pharmacy, Sungkyunkwan University, Suwon 16419, Republic of Korea. [2]College of Pharmacy and Research Institute for Drug Development, Pusan National University, Busan 46241, Republic of Korea. [3]Anti-infectives from Microbiota Helmholtz Institute for Pharmaceutical Research Saarland (HIPS) Campus E8.1, 66123 Saarbrücken, Germany. [4]Chemical Biology of Microbe-Host Interactions Leibniz institute for Natural Product Research and Infection Biology – Hans-Knöll-Institute (HKI), Beutenbergstraße 11a, 07745 Jena, Germany. [5]Department of Integrative Biological Sciences and Industry, Sejong University, Seoul 05006, Republic of Korea. [6]Saarland University, Inorganic Solid-State Chemistry, Campus, Building C4 1, 66123 Saarbrücken, Germany. [7]Department of Biochemistry, Genetics and Microbiology, Forestry and Agricultural Biotechnology Institute (FABI), University of Pretoria, Pretoria, South Africa. [8]Saarland University, 66123 Saarbrücken, Germany. [9]These authors contributed equally: Seoung Rak Lee, Marie Dayras.[10]These authors jointly supervised this work: Christine Beemelmanns, Ki Hyun Kim. ✉e-mail: christine.beemelmanns@helmholtz-hips.de; khkim83@skku.edu

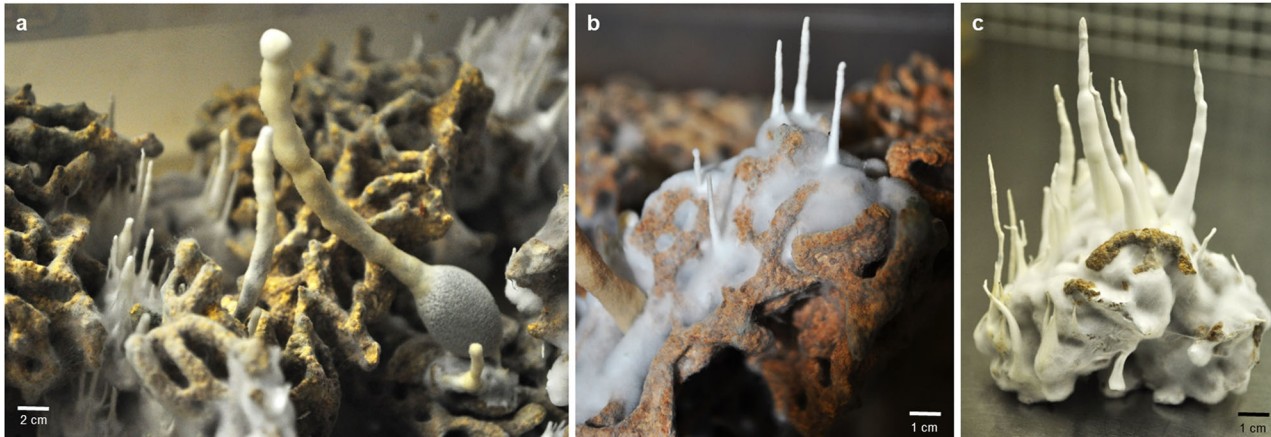

**Fig. 1 | Fungus comb overgrown by termite-associated *Xylaria*. a–c** Fungus comb derived from *Macrotermes natalensis* colonies overgrown by termite-associated *Xylaria* strains seven days after incubation in the absence of termites.

In our recent investigation, we delved into the genomic and biochemical foundations of the distinctive co-evolved antagonistic behavior and substrate specialization of termite-associated strains[10], and posited that the secreted antimicrobials could play a role in outcompeting the fungal mutualist once vegetative mycelium emerges[11–15]. Transcriptomic and metabolomic studies indeed showed that metabolite production is triggered and potentially diversified when termite-associated strains were cultivated on comb-mimicking growth conditions and in fungus-fungus co-cultivations against the fungal cultivar[10,16]. Furthermore, the chemical nature of secreted metabolites differed across the fungal isolates, which was also reflected in the isolate-specific genetically encoded biosynthetic repertoire. Especially, a biosynthetic gene cluster (BGC) region assigned to the cytosporin/xylasporin family caught our attention as the core BGC was encoded in the genome of four out of seven different termite-associated isolates, but variations in the abundance and arrangement of accessory genes was detectable. While genes coding for a short chain oxidoreductase (pxB; SDR) and a cupin protein (pxF) were lacking in the genomes of isolate X187, Mn153, and Mn132, the additional SnoaL-like polyketide cyclase (pxP) was present in X187, Mn153 and Mn132, and only lacking in isolate X802[11]. Therefore, we postulated the existence of a previously undiscovered level of structural diversity within the xylasporin/cytosporin family produced by strain X802, in contrast to those isolated from X187[10]. To further solidify the hypothesis and clarify the chemical nature of the cytosporin/xylasporins produced by the other termite-associated fungal isolates, we re-investigated the secondary metabolome of the model strain *Xylaria* sp. X802[8,11] using eco-mimetic cultivation conditions and a comparative HRMS$^2$-based metabolomic approach.

Here, we report the isolation and characterization of six yet unreported polyketide-terpene hybrid metabolites named xylasporins A-F (**1–6**) from *Xylaria* sp. X802 and the full structure determination of previously identified xylasporin I (**8**) from *Xylaria* sp. X187. Due to the structural complexity of this compound class, we pursued a comprehensive comparative NMR study coupled with chemical-computational calculation of more than 50 possible stereoisomers to enable the deduction of its relative and absolute stereochemistry. Furthermore, we were able to obtain single x-ray data of the commonly co-secreted polyketide-NRPS hybrid metabolite, xylacremolide B (**9**), enabling a revision of its stereochemistry.[14] Our comparative genomic studies also allowed deducing the putative polyketide-terpene-based biosynthetic origin.

## Results and discussion
### Cultivation and comparative analysis
To study if ecomimetic conditions could trigger the production of the cytosporin/xylasporin-specific features, *Xylaria* sp. X802[8,11] was co-cultivated with the fungal mutualist *Termitomyces* sp. T153 (7–14 days,

30 °C) (Fig. 2a)[10]. Culture extracts were analyzed for (induced) metabolite production by comparing the metabolome of the interaction zone (ZOI) to metabolites secreted from an axenic agar culture grown on potato-dextrose agar (PDA) and previous experimental set-ups[10] using liquid chromatography (LC) coupled with tandem electron-spray ionization (ESI)-HRMS/MS analyses. Obtained MS$^2$-data was dereplicated using the Global Natural Product Social Molecular Networking Web platform (GNPS) (Supplementary Fig. S1)[17]. The comparative molecular network revealed several molecular ion clusters containing more than ten different nodes, including a cluster putatively assigned to diketopiperazines, a cluster assigned to previously identified pseudoxylallemycins (*m/z* 549.322, 617.456, 619.262, and 685.394)[11], and a cluster assigned to cytochalasin derivatives (*m/z* 524.263) (Supplementary Fig. S2)[11]. However, the MS$^2$-cluster that was tentatively assigned to the cytosporin/xylasporin family from *Xylaria* sp. X802 included several spectral nodes with yet unknown molecular ion features, and which differed from those observed in the GNPS-subcluster of isolate X187 (Fig. 2b)[10].

We then investigated the abundances of cytosporin/xylasporin-specific features within culture extracts obtained from cultures grown on rice-sawdust medium or on glass beads (Fig. 2c, d)[10]. While the cytosporin/xylasporin-specific molecular ion features were detectable from *Xylaria* sp. X802 within all different cultivation conditions (Fig. 2e), some features were only detectable in samples derived from the interaction zone of a co-culture experiment (Fig. 2b). In addition, the intensity of each molecular ion feature was depended on the substrate provided for growth and was indicative for subtle transcriptional regulations of tailoring enzymatic activity as indicated by prior studies[10].

### Structural elucidation of xylasporin derivatives
We then pursued the isolation of the unknown xylasporin-specific chemical features from isolate X802[8,10,11]. For this, *Xylaria* sp. X802 was grown for 14 days on PDA plates and mycelium-covered plates were then extracted with methanol to yield an enriched crude extract. Compounds were purified by MS- and UV-guided reverse-phase chromatography. Overall, we were able to isolate seven metabolites. Comparative NMR analysis of compounds **1–6** uncovered a similar 1D and 2D NMR pattern, which indicated towards a shared bicyclic, and oxidized core skeleton (Tables S1 and S2). In addition, a known biosynthetic precursor, 1-hydroxy-2-hydroxymethyl-3-pent-1-enylbenzene (**7**) was characterized according to previous data.

The molecular formula of compound **1**, $C_{18}H_{26}O_7$, was deduced from HRESI( + )MS data [*m/z* 709.3430 [2 M + H]$^+$, calcd. for $C_{36}H_{53}O_{14}$, 709.3435]. The $^1$H NMR data of **1** showed the presence of signals for three methyl groups, three methylene units, one oxygenated methylene, four oxygenated methines, and two olefinic protons (Tables S1 and S2). HSQC and HMBC spectra revealed a total of 18 carbon signals attributable to three

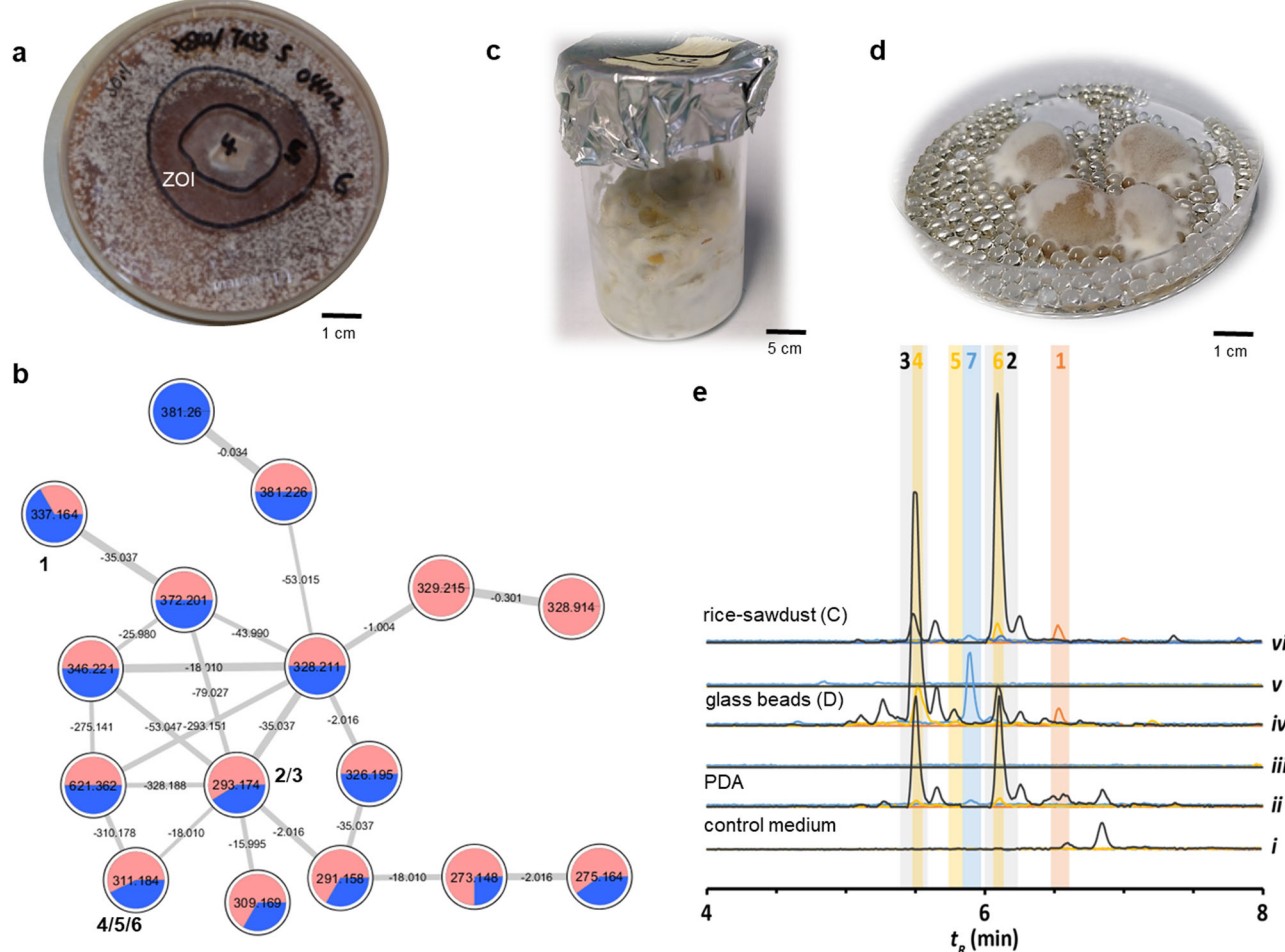

**Fig. 2 | Metabolomic analysis of *Xylaria* sp.** X802 grown on different media and co-culture conditions. **a** Co-cultures of *Xylaria* sp. X802 versus *Termitomyces* sp. T153 on PDA/soil medium; (**b**) Expanded GNPS-based molecular network cluster of xylasporin-based molecular ion features. Features that were only present in the zone of inhibition (ZOI) are of pink color and those from axenic growth of X802 are colored in blue (**1**, *m/z* 337.164 [M-H₂O + H]⁺, **2/3**, *m/z* 293.174 [M-H₂O + H]⁺, and **4/5/6** *m/z* 311.184 [M-H₂O + H]⁺). **c** Culture of *Xylaria* sp. X802 on rice-sawdust medium, and (**d**) glass beads with liquid PDB. **e** Selected ion chromatogram of **1** (*m/z* = 337.1636 [M-H₂O + H]⁺), **2/3** (*m/z* = 293.1740 [M-H₂O + H]⁺), **4/5/6** (*m/z* = 311.1847 [M-H₂O + H]⁺) and **7** (*m/z* = 193.1223 [M + H]⁺) with extracts derived from (ii) agar plate cultivation, (iv) glass beads cultivation with liquid PDB and (vi) rice-sawdust medium of three weeks old *Xylaria* sp. X802. The corresponding negative controls without X802 are shown below each lane (i, iii and v), respectively.

methyls, four methylenes, six methines, four quaternary carbons, and one carbonyl-like carbon (Table S1). Interpretation of 2D NMR spectra allowed the determination of planar structure of **1**, with notable HMBC correlations between H-7/C-14, and H-6/C-19 and H-7/C-19 showing the presence of a cyclic carbonate group, as well as COSY correlations of an aliphatic spin system (Fig. 3a). Overall compound **1** showed similarities to cytosporin E featuring a pentenyl chain at C-8 compared to a heptenyl chain in cytosporin E (Fig. 3c)[18]. The relative configuration of **1** was determined by vicinal ¹H-¹H coupling constants (³$J_{HH}$) and ROESY correlations of H-3/H₃-12, H-10/H₃-11, H₃-11/H-4α, and H-6/H-4β indicating the equatorial orientation of the hydroxyl group at C-3 and the axial orientation of H-6, H-7, and H-10 (Fig. 3a). As the hexahydrobenzopyrane moiety of compound **1** showed NMR chemical shifts very close to those of cystoporin E, we deduced that these compounds share the same absolute configuration: 3 *S*, 5 *R*, 6 *R*, 7 *R*, and 10 *S* (Supplementary Table S3, Supplementary Figs. S3–S8).To solidify our structural assumptions, we also calculated the ¹H and ¹³C shifts using optimized settings for Gauge-including atomic orbital (GIAO) NMR chemical shifts calculations, which was followed by DP4+ probability analysis[19] of overall 50 stereoisomers of the three structure types that encompass compound **1–6** (Supplementary Tables S11–S22). This effort allowed us to confirm the *cis*-orientation of the angular substituents at C-5 and C-10 of compound **1** (Supplementary Tables S4 and S5, Supplementary Fig. S9).

Additionally, we compared the experimental ECD spectrum of **1**, which exhibits two positive Cotton effects at 228 nm and 265 nm, with the calculated ECD spectra for the most stable conformers of the two possible enantiomers of **1** [**1a** (3*S*,5*R*,6*R*,7*R*,10*S*) and **1b** (3*R*,5*S*,6*R*,7*S*,10*R*)], which in summary allowed us to deduce the absolute configuration for compound **1** (3*S*,5*R*,6*R*,7*R*,10*S*) (Fig. 3b). Thus, we named compound **1** xylasporin A due to its structural resemblance to the cytosporin family[20–29], Here, it was intriguing to note that cyclic-carbonate patterns are rarely found in natural products, and have so far predominantly been reported from the marine-derived fungus *Peroneutypa scoparia* (previously termed *Eutypella scoparia*)[20], *Aspergillus* sp. PSU-RSPG185 (aspergillusols)[23], and *Phoma* sp. (phomoxins) (Fig. 3c)[30].

The molecular formula of xylasporins **2** and **3** was determined as C₁₇H₂₆O₅ based on the HRESI(+)MS analyses, which showed pseudo-molecular ions peaks at *m/z* 293.1742 [M-H₂O + H]⁺ (calcd. for C₁₇H₂₅O₄, 293.1753) for **2** and *m/z* 621.3627 [2 M + H]⁺ (calcd. for C₃₄H₅₃O₁₀, 621.3639) for **3**, indicating that these compounds are likely structural isomers. The 1D and 2D NMR spectra of **2** and **3** exhibited some similarities with those signals assigned to the core structure of **1**. However, signals for the carbamate group were lacking and chemical shift patterns at position C-5, C-6, and C-7 were suggestive for the presence of an epoxide group at C-5 and C-6 (Fig. 4a, b, Tables S1 and S2).

**Fig. 3 | Structural analysis of xylasporin A. a** Planar structure of xylasporin A (**1**) and key 2D NMR correlations for compound **1**; (**b**) Experimental ECD spectrum for compound **1** and Boltzmann-averaged TD-DFT calculated ECD spectra for **1a** (3S,5R,6R,7R,10S), **1b** (3R,5S,6S,7S,10R); (**c**) Proposed absolute structure of xylasporin A (**1**) and reported cytosporin E. **d** Structures of known carbamate-containing fungal metabolites phomoxins and aspergillusols sharing similar structural and biosynthetic features.

**Fig. 4 | Structural analysis of xylasporin B and C.**
**a** Proposed structure of xylasporin B (**2**), structure with |Δδ$_{C(2\text{-cystoporin D})}$| values in ppm and 3D depiction showing ROESY correlations; (**b**) Proposed structure of xylasporin C (**3**), structure with |Δδ$_{C(3\text{-cystoporin D})}$| values and 3D depiction showing ROESY correlations; (**c**) Structures of reported natural products with similar core scaffolds used for comparison.

Comparison with literature data suggested that compounds **2** and **3** had a similar planar structure to cytosporins D and M, but carried a pentenyl chain at C-8 in contrast to a heptenyl chain in cytosporins D and M (Fig. 4c)[18,27]. Differences in chemical shifts and $^3J_{HH}$ coupling constants for H$_2$-4, as well as key ROESY correlations supported the notion that compounds **2** and **3** share the same relative configurations 3 S*, 5 R* and 6 S* with an inversion of relative configurations for C-7 and C-10 (**2**: 7 R*, 10 S*; **3**: 7 S*, 10 R*) (Fig. 4a, b). Comparison of $^1$H and $^{13}$C NMR chemical shifts of the hexahydrobenzopyrane moieties of compound **2** and cytosporin D led us to deduce the putative absolute configuration of **2** as 3 S, 5 R, 6 S, 7 R, and 10 S [(X̄(|Δδ$_{H(2\text{-cystoporin D})}$|)) = 0.04 ppm, s = Δδ$_{H(2\text{-cystoporin D})}$ = 0.05 ppm; X̄(|Δδ$_{C(2\text{-cystoporin D})}$|) = 0.24 ppm, s = Δδ$_{C(2\text{-cystoporin D})}$ = 0.15 ppm)] (Fig. 4a, Supplementary Tables S6–S9, Supplementary Figs. S10–S22)[18]. These deductions were solidified by GIAO NMR chemical shifts calculation followed by DP4+ probability, which additionally allowed us to propose the relative configuration for

xylasporin C (**3**) as 3S*,5R*,6S*,7S*, and 10R* (Supplementary Table S10–S13, Supplementary Figs. S22 and S23).

The molecular formulas of xylasporins D (**4**), E (**5**), and F (**6**) were deduced from the sodium-adducted ion peak at m/z 351.1784 [M+Na]$^+$ (calcd. for C$_{17}$H$_{28}$O$_6$Na, 351.1784) from HRESI(+)MS data of **4**, and from the pseudo-molecular ions peaks at m/z 657.3850 [2 M + H]$^+$ and m/z 657.3855 [2 M + H]$^+$ (calcd. for C$_{34}$H$_{57}$O$_{12}$, 657.3850) from HRESI( + )MS data of **5** and **6**.

Interpretation of the $^1$H and $^{13}$C NMR spectra of these compounds revealed again the hexahydrobenzopyrane scaffold with chemical shift differences at C-5, C-6, and C-7 compared to **1**, which suggested the presence of hydroxyl groups at C-5, C-6, and C-7. COSY and key HMBC correlation analyses allowed deducing the planar structures of **4**, **5**, and **6**, resembling that of cytosporin L, which includes a pentenyl chain at C-8 compared to a heptenyl chain in cytosporin L (Fig. 5,)[3]. Possible relative configurations of C-3, C-5, C-6, C-7, and C-10 of **4**

**Fig. 5 | Structural analysis of xylasporin D-F.**
**a** Proposed structure of xylasporins D-F (**4-6**), compound **7**, and of similar reported compound, cytosporin L; (**b**) Key 2D NMR correlations found for compounds **4**, **5** and **6**.

were deduced from key ROESY correlations (Fig. 5b, Supplementary Figs. S25–S43) and GIAO NMR chemical shifts calculations of a reduced set of possible stereoisomers followed by DP4+ probability all together supported the proposed configuration of 3 $S^*$, 5 $S^*$, 6 $R^*$, 7 $R^*$, and 10 $R^*$ for compound **4** (Supplementary Tables S14 and S15, Supplementary Fig. S30).

We then compared the [1]H and [13]C chemical shifts for compound **5** and **6** and found that the chemical shift values of C-6 and C-7, and the vicinal coupling constant between the protons H-6 and H-7 (**4**: 6.5 Hz *vs.* **5** and **6**: 4.5 Hz) in relation to those values obtained for compound **4** were suggestive for an *S*-stereochemistry at position C-6 and C-7. This deduction was also supported by key ROESY correlations (H-6/H-10 for **5** and H-6/H-4β for **6**). Stereochemistry at position C-10 was deduced for both compounds from signatory [13]C chemical shift patterns and ROESY correlations (**5**: H-10/H-3, H-3/H₃-12, and H₃-12/H-4α; **6**: H-10/H-4α and H-4α/H₃-11) (Fig. 5b). The stereochemical configuration of C-5 in compound **5** remained an open question as the experimental and calculated NMR data resulting of DP4+ probability of 78% (Supplementary Tables S16 and S17, Supplementary Fig. S37). While still ambiguous, we propose that ROESY correlations between H-10 and H-3, and H-10 and H-6 are suggestive for a 5 *S* configuration for compound **5**. In contrast, the stereochemistry at C-5 for compound **6**, was determined based on the key H-10/H-4α ROESY correlation and unambiguous DP4+ analysis of 100% (Fig. 5B, Supplementary Tables S18 and S19, Supplementary Fig. S44). Overall, the combination of NMR interpretation and calculation allowed us to proposed the relative configuration for **5** as 3 $S^*$,5 $S^*$,6 $S^*$,7 $S^*$,10 $R^*$ and for **6** as 3 $S^*$,5 $R^*$,6 $S^*$,7 $S^*$,10 $S^*$.

Finally, we also compared the [1]H and [13]C shifts using optimized settings for GIAO-NMR chemical shifts calculations of several stereoisomers across each structure type (Tables S4–S5, S10–S13, S14–S19). Overall, we were able to deduce patterns ("rule of thumbs") that correlated to the stereochemistry of C-3 and C-10 and the [13]C chemical shifts of C-11 and C-12. In case the same stereochemistry (*S*, *S* or *R*, *R*) was observed for C-3 and C-10, the [13]C chemical shifts of C-11 and C-12 differed by more than 10 ppm.

In case the stereochemistry of C-3 and C-10 were inverted (*R*, *S* or *S*, *R*), the [13]C chemical shift values for C-11 and C-12 differed less than 3 ppm.

## Relative structure elucidation of xylasporin I and structure revision of xylacremolide B from isolate X187

Prior comparative genomic and metabolomic analyses proposed that *Xylaria* strains carrying *fog*-like gene clusters should produce xylasporin-like metabolites, which led to first isolation efforts and deduction of the planar structure of two congeners xylasporin H and I from strain X187[10]. However, due to the light- and acid-sensitive chemical features of the conjugated polyene system, only limited analytical data was obtained leaving the relative stereochemistry of these congeners unassigned. Building up on the acquired analytical and calculated NMR data of the xylasporin-compound class within this study, we herein addressed again the intrinsic challenge of assigning the relative stereochemistry to isolates of this compound family. For re-isolation, agar plates covered with fungal mycelium of *Xylaria* sp. X187 were used, which yielded after ethyl acetated-based extraction, xylasporin I (**8**) in 1.2 mg alongside with the co-secreted natural products xylacremolides A and B (**9**) (Fig. 6)[10,14]. Despite the instability of xylasporin I (**8**), we were able to obtain this time sufficient quality of 2D-NMR data for structural and computational analysis (Supplementary Table S20, Supplementary Figs. S48–54). 2D NMR analysis confirmed that xylasporin I (**8**) shares the core structure and chain length with xylasporins B (**2**) and C (**3**) produced by *Xylaria* sp. X802; however, it was distinguished by a pentdienyl chain instead of the heptenyl chain present in cytosporins D and M[18,27]. Comparative NMR studies and GIAO NMR chemical shifts calculations using optimized settings (Supplementary Tables S21–S22, Supplementary Fig. S55) allowed us to propose the relative stereochemistry for xylasporin I (**8**) as 7 $R^*$, 8 $S^*$, 9 $R^*$, 11 $S^*$, 15 $R^*$ (Fig. 6a).

In light of our efforts to obtain single crystals for any of the secondary metabolites isolated from strains X802 and X187 to validate the deductions from chemical shift calculations, only purified xylacremolide B (**9**) crystallized from a mixture of methanol and water (Fig. 6b, Supplementary Table S24, Supplementary Fig. S61).

**Fig. 6 | Structural assignment of xylasporin I and revision of chemical structure of xylacremolide B. a** Chemical structure of xylasporin I (**8**) and its 3D depiction showing ROESY correlations. **b** Revised chemical structure of xylacremolide B (**9**), crystal structure of xylacremolide B (**9**) shown as ORTEP plot of with displacement ellipsoids of non-hydrogen atoms drawn at 50% probability level and chemical structure of xylacremolide A[14].

In combination with our previous reports, which included the determination of the relative stereochemistry (D-Phe and L-Pro) of xylacremolides A and B (**9**) using Marfey's method and a single crystal X-ray analysis of xylacremolide A, we were able to serendipitously revise the chemical structure of xylacremolide B (**9**) (Fig. 6b)[14]. Single X-ray analysis of **9** uncovered two important features: firstly, the observed orientation of the 2-CH₃ towards the aromatic ring of Phe indeed leads to a direct exposure to the ring current, likely causing the significant upfield-shift of the chemical shift assigned to 2-CH₃[14]. Secondly, while previous NOESY interpretation suggested a 2 $S$,3 $R$,5$R$-configuration of the PKS-derived hydroxylated alkyl chain, single X-ray analysis uncovered a 2 $R$,3 $R$,5$R$-configuration, and thus let to a revision of the relative and absolute configuration of xylacremolide B (**9**).

**Biosynthetic considerations**

Cytosporins and xylasporins share the same planar hexahydrobenzopyran backbone, with xylasporins carrying a pentenyl chain at C-8 compared to a heptenyl chain in cytosporins. As the planar hexahydrobenzopyran backbone suggested a mixed polyketide and terpene-type biosynthetic origin[18–31], we revisited our previous comparative genome analysis of *Xylaria* genomes[10], which reported four strains (X802, Mn132, Mn153, and X187) to encode a cluster region bearing resemblance to the *fog* BGC (*Aspergillus ruber*)[4] responsible for the biosynthesis of glaucin-type of scaffolds. The locus within the genome of Xylaria sp. X802 was assigned the *px* gene cluster (BGC) region (Fig. 7, Supplementary Table S25), which encompasses a highly-reducing poly ketide synthase PKS (HR-PKS) *pxA* (KS-AT-DH-ER-KR-ACP), an ABBA type aromatic prenyltransferase *pxH*, three short-chain dehydrogenases/reductases (SDR) *pxB*/*pxD*/*pxJ*, two cupins *pxC*/*pxF*, a cytochrome p450 monooxygenases (CYP) *pxE*, two FAD-dependent oxidoreductases *pxG*/*pxI*, two fungal transcription factors *pxK*/*pxM*, a major-facilitator superfamily (MFS) transporter *pxL*, a phosphopantetheinyl transferase *pxN* and a thioesterase domain containing enzyme *pxO*[32].

A detailed in silico of the *px* BCG uncovered that seven of the encoded proteins (PxA-E and PxG-I) showed high similarity (42–86% similar amino acid sequence; Supplementary Table S20) to the characterized FogA-F from *Aspergillus ruber* sp. CBS 135680, which is responsible for the biosynthesis of the prenylated salicylic acid derivative flavoglaucin[31]. Homologs gene sequences were also previously identified in other ascomycetes like *Trichoderma virens* sp. Gv29-8 (61–80%)[33] and *Neurospora crassa* sp. OR74A (47–76%)[34], encoding for the biosynthesis of the prenylated PKS-based salicylic acid derivatives sordarial and trichoxide, respectively. Based on

these homologies, an iterative incorporation of five malonyl-CoA onto an acetyl-CoA starter unit by the HRPKS PxA was proposed to yield ACP-bound intermediate I. Here, it is important to point out that xylasporins A-F (**1–6**) are characterized by a pentenyl chain at C-8 compared to a heptenyl chain in cytosporins, which suggested yet unknown structural and functional difference in the iterative mode of the HRPKS PxA compared to the fogA-PKS reported in *Aspergillus*.

The strain-specific modifications of the polyketide chain should be realized by the KR, DH, and ER domains of the HR-PKS PxA as well as the short-chain dehydrogenases/reductase PxB and cupin PxC, similar to mechanisms that have been shown for flavoglaucin biosynthesis in *A. ruber*[31]. Release of the polyketide II is proposed to be catalyzed by SDR PxD or the putative *trans*-acting TE domain containing protein PxO leading to the production of the detectable precursor metabolite **7**. Hydroxylation of compound **7** at position C-3 by PxE and prenylation at C-5 by PxH could lead to the synthesis of a flavoglaucin congener (IV). The double bond at the prenyl group should then be epoxidized by either the putative cytochrome P450 monooxygenases PxF or PxG. The chromene core structure of VI is likely formed by spontaneous ring closure leading to two distinct stereoisomers ($m/z = 275.164$ [M-H₂O + H]⁺). Subsequent oxidation of the alcohol at C-7 by either PxI or PxJ might form an unsaturated enone VIII, while a second epoxidation by PxF or PxG could catalyze the formation of **2** and **3**. Enzymatic or spontaneous hydrolysis of the oxyrane ring and reduction of the carbonyl functionality should provide the three isomers **4–6**, but the enzymatic basis for the regioselective introduction of the cyclic carbonate moiety still remains enigmatic.

**Antibacterial activity studies**

Cytosporins A-C, isolated from an endophytic strain of the taxonomically related fungus *Cytospora*[5], were reported to act as angiotensin II binding inhibitor, while cytosporin L was shown to have activity against the bacteria *Micrococcus lysodeikticus* and *Enterobacter aerogenes* with MIC values of 3.12 µM[18]. However, cytosporin D and E were reported to have neither antibacterial nor antifungal activity against *Staphylococcus aureus*, *Escherichia coli*, and *Candida albicans*[23,27].

Indeed, when we evaluated the antibacterial activity of stable xylasporins derivatives A-F (**1–6**) against a panel of bacterial test strains, neither of the derivatives exhibited antibacterial activity. As xylasporins A-F (**1–6**) were detectable in the ZOI of the fungus-fungus co-culture (Fig. 2), their antifungal activity against *A. nidulans* RMS011 as test strain was evaluated. However, only very mild antifungal activity in form of spore germination inhibition was observed for xylasporins B (**2**) and C (**3**) to a similar extend as

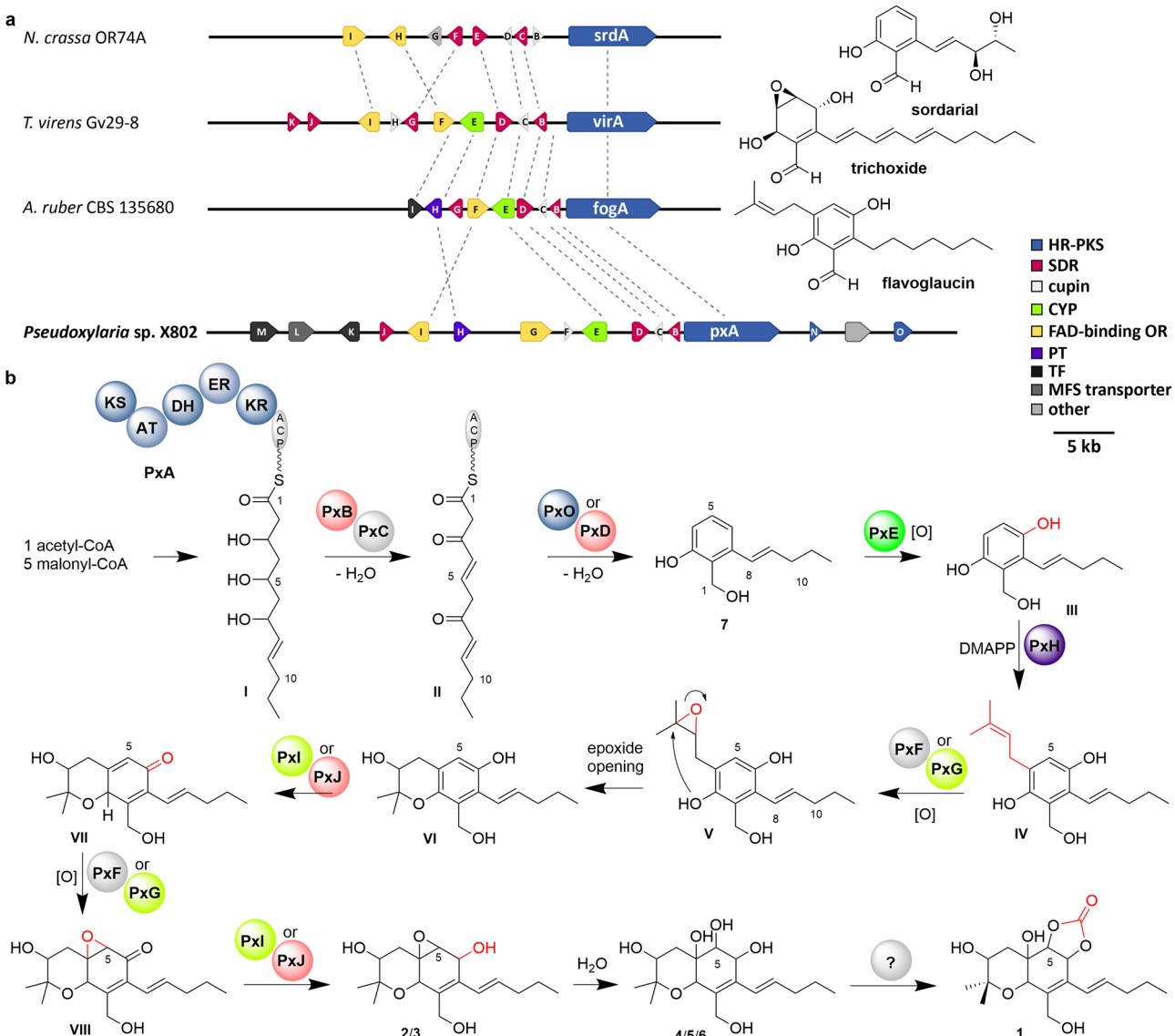

**Fig. 7 | Comparative gene cluster analysis of the *px* cluster. a** Schematic depiction of the *px* BGC in *Xylaria* sp. X802 and comparison to homologs BGCs from the biosynthesis of flavoglaucin (*Aspergillus ruber* CBS 135680), trichoxide (*Trichoderma virens* sp. Gv29-8) and sordarial (*Neurospora crassa* sp. OR74A). Shown are genes coding for highly-reducing polyketide synthases (blue; HR-PKS), short-chain dehydrogenases/reductases (red; SDR), cupins (silver), cytochrome p450 monooxygenases (green; CYP), flavin dependent oxidoreductases (yellow; OR), aromatic prenyltransferases (purple; PT), transcription factors (black) and major facilitator superfamily transporters (dark gray; MFS). Non-related genes are shown in light gray. **b** Putative biosynthetic assembly line yielding isolated biosynthetic precursor **7**, which is likely biotransformed through a series of oxidative transformation steps to yield epoxidized xylasporins of type **2/3**, tetradiols **4-6** and in a yet unidentified step carbonate **1**.

the co-isolated 19,20-epoxycytochalasin Q and pseudoxylallemycin A (Supplementary Fig. S62)[11].

## Conclusion

Members of the termite-associated subgenus *of Xylaria* sp. are substrate-specialized and emerge from abandoned fungus comb material of fungus-growing termites. Building on the discovery that termite-associated strains inhibit the growth of the fungal crop *Termitomyces* assisted by its biosynthetic capabilities to produce secondary metabolites, we substantiated the hypothesis by conducting experiments under eco-mimetic growth conditions and employing a comparative HRMS²-based metabolomic approach. Dereplication of metabolic features, whose abundances were dependent on the provide growth conditions, led to the discovery of six yet unreported secondary metabolites of PKS-terpenoid origin from strain X802, which we named xylasporins A-F (**1–6**) due to their hexahydrobenzopyran backbone and structural resemblance to cytosporins, and the reisolation of the instable

xylasporin I (**8**) from strain X187. GIAO NMR calculations, combined with 2D NMR interpretation were used to complete the determination of their relative structures, which was in agreement with oxygenation pattern, and the unsaturation degree within the alkenyl chain were deduced from the predicted PKS domain arrangement. Targeted analysis allowed to map the abundance of xylasporins across termite-associated *Xylaria* isolates. While derivatives featuring a pentadienyl chain at C-8 were predominately produced by *Xylaria* sp. X187, *Xylaria* sp. X802 was found to produce only derivatives featuring a pentenyl chain at C-8, and thus differentiates those from cytosporins, which are characterized by a heptenyl chain and are produced by other fungal lineages. The weak antimicrobial inhibitory activity of only few of the isolated secondary metabolites supported our previous conclusion that termite-associated *Xylaria* might have adapted to such comb-specific stressors by reducing and specializing the secondary metabolome to minimize triggers that could stimulate alarm responses of the fungal mutualist and termites. Our combined metabolomic, structural

analytical, and genomic study now allows to more accurately map the abundance of the PKS-terpenoid derived cytosporin/xylasporin across fungal genera and predict the structural features and relative stereochemistry based on our extensive GIAO NMR calculations encompassing calculated NMR data of more than 50 stereoisomers.

## Method Section

### General experimental procedures

**Chemicals.** All media, solvents, and, chemicals were purchased as follows: methanol, dichloromethane, ethyl acetate, acetonitrile (Th. Geyer, Renningen); water for analytical and preparative HPLC (Millipore, Germany), formic acid (Carl Roth, Germany), MeOH-$d_4$, CDCl$_3$ (Carl Roth, Germany), media ingredients (Carl Roth, Germany). Merck precoated silica gel F254 plates and RP-18 F254s plates were used for thin layer chromatography (TLC). Spots were detected on TLC under UV light or by heating after spraying with anisaldehyde-sulfuric acid.

**Strains and culture conditions.** A pre-culture of *Xylaria* sp. X802[5,11,14] was prepared by inoculating 200 mL PDB with 1–2 agar cubes of a 2-week old *Xylaria* sp. X802 culture and incubated for 2 weeks at 30 °C and 150 rpm. Afterwards, the pre-culture was used to inoculate 20 PDA plates (92 × 16 mm), petri dishes filled with glass beads containing 10 mL liquid PDB and 10 mL rice-sawdust medium mixed in a 1:1 (v/v) ratio. The cultures incubated for 3 weeks at RT. Media served as negative controls. The cultures from liquid PDB and rice-sawdust medium were extracted with each 50 mL ethyl acetate overnight. Residues were combined, evaporated under reduced pressure, dissolved in MeOH and analysed using UPLC-ESI-HRMS.

**Co-cultivation studies.** *Xylaria* sp. X802 and *Termitoymces* sp. T153 were co-cultivated on PDA plate for three weeks as inoculum[10,11]. The mycelium of strain T153 was scratched from the surface of PDA plate and collected in sterile PBS (20 mL) to make nearly homogeneous mycelium suspension by vortex. Due to the different growth rate of both fungal strains, two different co-culture growth conditions on standard petri dish (9 cm diameter) containing 20 mL of 1/3 PDA medium were set-up. *Method A*: *Xylaria* sp. X802 and *Termitoymces* sp. T153 were inoculated simultaneously: one mycelium containing agar plug of X802 (1 × 1 cm) was placed in the middle of petri dish. Then 200 μL of T153 mycelium suspension was placed on the empty space of the same petri discs and evenly distributed by gentle spreading. *Method B*: *Xylaria* sp. X802 was grown for one week until vegetative mycelium of *Xylaria* sp. X802 was visible. Then *Termitoymces* sp. T153 mycelium suspension was introduced close to the X802 colony. Axenic cultures of *Xylaria* sp. X802 and *Termitoymces* sp. T153 were cultivated for the same time period as controls. To analyze the secreted metabolites within the zone of inhibition, the interaction zone from two-week co-cultivation plate was separated, cut into small pieces and extracted by 20 mL of dichloromethane (DCM) overnight. Similarly, mycelium of *Xylaria* sp. X802 and *Termitoymces* sp. T153 were separated, cut into small pieces and extracted by 20 mL of DCM for overnight respectively. Controls (*Xylaria* sp. X802 and *Termitomyces* sp. T153) were extracted in the same way. The DCM extract were collected by filtration and concentrated under reduced pressure using a SpeedVac. The resultant DCM extract were dissolved into LCMS grade MeCN as 0.1 mg/mL and submitted to UPLC-ESI-HRMS analysis.

**HR-MS.** High resolution ultra-high performance liquid chromatography electrospray ionization mass spectrometry (UPLC-ESI-HRMS) measurements were performed on a Dionex Ultimate3000 system coupled with a Luna Omega C18 column (100 × 2.1 mm, particle size 1.6 μm, pore diameter 100 Å, Phenomenex) combined with Q-Exactive Plus mass spectrometer (Thermo Scientific) equipped with an electrospray ion (HESI) source. Column oven was set to 40 °C; scan range of full MS was set to *m/z* 150 to 2000 with resolution of 70,000 and AGC target 3e6 and

maximum IT 100 ms under positive and negative mode with centroid data type. MS$^2$ was performed to choose top10 intensive ions under positive mode with resolution of 17,500 and AGC target 1e5 and maximum IT 50 ms and (N)CE (20/30/40) with centroid data type. The spray voltage (+) was set to 4000 volt, and (−) was set to 3300 volt. The capillary temperature (±) was set to 340 °C and probe heater temperature (±) was set to 200 °C. The sheath gas flow (±) was set to 35 L/min and Aux gas flow (±) to 5 L/min. Max spray current (+) and (−) was set to 100 volt. S-Lens RF level was set to 50.

**GNPS analysis.** Culture extracts were submitted to LC-ESI-HRMS metabolomics analysis under standard condition. Metabolites were separated under the gradient: 0–0.5 min, 5% B; 0.5–18 min, 5%–97% B; 18–23 min, 97% B; 23–25 min, 97%–5% B; 25–30 min, 5% B (A: 0.1% FA; B: MeCN with 0.1% FA), with flow rate of 0.3 mL/min and injection volume is 5 μL. Metabolomics raw data acquired on a Thermo QExactive Plus mass spectrometer was converted to 32-bit mzXML files using MSConvert GUI (ProteoWizard), in order to generate a mass spectral molecular networking using the global natural products social molecular networking platform (GNPS, https://gnps.ucsd.edu)[6]. Data analysis used default parameters, except for the cosine threshold, set to 0.7, minimum matched fragment ions of 4, network TopK 10, and for the tolerances of the precursor- and fragment ion masses, both set to 0.02 Da. The mass spectral network was assembled and visualized using Cytoscape[35].

**Analytical and preparative HPLC.** High-performance liquid chromatography (HPLC) were performed on a Waters 1525 Binary HPLC pump with a Waters 996 Photodiode Array Detector (Waters Corporation, Milford, CT, USA). Semi-preparative HPLC used a Shimadzu Prominence HPLC System with SPD-20A/20AV Series Prominence HPLC UV-Vis Detectors (Shimadzu, Tokyo, Japan).

**NMR.** NMR spectra, including $^1$H-$^1$H COSY, HSQC, HMBC, and ROESY experiments, were carried out using a Varian UNITY INOVA 800 NMR spectrometer operating at 800 MHz ($^1$H) and 200 MHz ($^{13}$C) or a Bruker Ascend 700 at 700 MHz ($^1$H) and 175 MHz ($^{13}$C), with chemical shifts given in ppm (δ). MestReNova ver. 12.0.1 was used to process the data for the NMR spectra.

**Analytical characterization.** Experimental ECD spectra in MeOH were acquired in a quartz cuvette of 1 mm optical path length on a JASCO J-1500 spectropolarimeter (Tokyo, Japan). IR spectra were acquired on a Bruker IFS-66/S FT-IR spectrometer. Optical rotations were obtained utilizing a Jasco P-1020 polarimeter (Jasco, Easton, MD, USA).

### Crystallography data

**X-ray analysis.** The data set was collected using a Bruker D8 Venture diffractometer with a microfocus sealed tube and a Photon II detector. Monochromated Mo$_{K\alpha}$ radiation (λ = 0.71073) was used. Data were collected at 150(2) K and corrected for absorption effects using the multi-scan method. The structure was solved by direct methods using SHELXT[36] and was refined by full matrix least squares calculations on F$^2$ SHELXL2018[37] in the graphical user interface Shelxle[38].

**Refinement.** All non H-atoms were located in the electron density maps and refined anisotropically. C-bound H atoms were placed in positions of optimized geometry and treated as riding atoms. Their isotropic displacement parameters were coupled to the corresponding carrier atoms by a factor of 1.2 (CH, CH$_2$) or 1.5 (CH$_3$). The O3 and N2 bonded H-atoms H3O and H2N were located in the electron density maps. Their positional parameters were refined using isotropic displacement parameters which were set at 1.2 (N2) and 1.5 (O3) times the Ueq value of the parent atoms. Restraints of 0.88(0.01) Å and 0.84(0.01) Å were used for the N-H and O-H bond lengths, respectively.

Crystal data for **9**: $C_{23}H_{32}N_2O_5$, Mr = 416.50 gmol$^{-1}$, size 0.200 × 0.060 × 0.060 mm$^3$, orthorhombic, space group P 2$_1$ 2$_1$ 2$_1$, a = 9.2915(3), b = 10.7976(3), c = 22.0427(6) Å, α = 90, β = 90, γ = 90°, V = 2211.45(11) Å$^3$, T = 150 K, Z = 4, ρ$_{calcd.}$ = 1.251 gcm$^{-3}$, μ = 0.088 mm$^{-1}$, multi-scan, transmin: 0.706, transmax: 0.746, F(000) = 896, 72893 reflections in h(-11/11), k(-13/13), l(-28/28), measured in the range 2.100° ≤ Θ ≤ 27.114°, completeness Θ$_{max}$ = 100%, 4888 independent reflections, R$_{int}$ = 0.0521, 279 parameters, 2 restraints, R1$_{obs}$ = 0.0291, wR2$_{obs}$ = 0.0695, R1$_{all}$ = 0.0318, wR2$_{all}$ = 0.0718, GOOF = 1.055, Flack-parameter -0.0(2), largest difference peak and hole: 0.165 / -0.170 e Å$^{-3}$.

Crystallographic data can be found in (Table S24) and has been deposited at the Cambridge Crystallographic Data Centre under CCDC-2347670 (Supplementary Data 2). This data can be obtained free of charge via www.ccdc.cam.ac.uk/conts/retrieving.html (or from the Cambridge Crystallographic Data Centre, 12, Union Road, Cambridge CB2 1EZ, UK; fax: (+44) 1223-336-033; or deposit@ccdc.cam.ac.uk).

**Isolation of compounds 1-7 from Xylaria sp. X802.** A flask with 50 mL PDB was inoculated with 2 agar cubes of a 2-week old *Xylaria* sp. X802 culture. The culture was incubated for 2 weeks at 30 °C and 150 rpm. Afterwards, the pre-culture was used to inoculate 120 PDA plates (20 mL PDA, 92 × 16 mm), which were kept at 26 °C for 14 days in the dark. Agar plates were then cut into pieces, soaked in EtOAc overnight. The organic extract was filtered of and the agar pieces were subsequently soaked in MeOH overnight, and filtrated again. The organic extracts were combined and dried *under vacuo*. The fungus crude extract (9.5 g) was suspended in distilled water (700 mL) and successively solvent-partitioned with EtOAc (700 mL) three times, yielding 2.0 g of residue. The EtOAc-soluble fraction was loaded onto silica-gel column chromatography and fractionated using a gradient solvent system of $CH_2Cl_2$-MeOH (100:1–1:1, v/v) to give five fractions (A-E). The fraction B (479 mg) was fractionated by preparative reversed-phase HPLC (Phenomenex Luna C18, 250 × 21.2 mm i.d., 5 μm) using $CH_3CN$-$H_2O$ (1:9–1:0, v/v, gradient system, flow rate: 5 mL/min) to provide six subfractions (B1–B6). Subfraction B4 (56 mg) was isolated by a semi-preparative reversed-phase HPLC (Phenomenex Luna C18, 250 × 10.0 mm i.d., 5 μm) with 70% MeOH/$H_2O$ (isocratic system, flow rate: 2 mL/min) to yield compound **7** (1.7 mg, $t_R$ = 47.0 min). The fraction C (160 mg) was separated by preparative reversed-phase HPLC (Phenomenex Luna C18, 250 × 21.2 mm i.d., 5 μm) using $CH_3CN$-$H_2O$ (1:9–1:0, v/v, gradient system, flow rate: 5 mL/min) to yield five subfractions (C1–C5). Compound **1** (1.4 mg, $t_R$ = 24.0 min) was isolated from subfraction C3 (40 mg) by a semi-preparative reversed-phase HPLC (Phenomenex Luna C18, 250 × 10.0 mm i.d., 5 μm) with 40% MeOH/$H_2O$ (isocratic system, flow rate: 2 mL/min). Fraction C4 (20 mg) was purified by a semi-preparative reversed-phase HPLC (Phenomenex Luna C18, 250 × 10.0 mm i.d., 5 μm) with 45% MeOH/$H_2O$ (isocratic system, flow rate: 2 mL/min) to afford compound **2** (2.2 mg, $t_R$ = 30.0 min). The fraction D (101 mg) was subjected onto a preparative reversed-phase HPLC (Phenomenex Luna C18, 250 × 21.2 mm i.d., 5 μm) and separated by $CH_3CN$-$H_2O$ (2:8–1:0, v/v, gradient system, flow rate: 5 mL/min) to yield five subfractions (D1–D5). Subfraction D2 (18 mg) was isolated by a semi-preparative reversed-phase HPLC (Phenomenex Luna C18, 250 × 10.0 mm i.d., 5 μm) utilizing 60% MeOH/$H_2O$ (isocratic system, flow rate: 2 mL/min) to give compounds **4** (1.2 mg, $t_R$ = 31.0 min), **3** (1.5 mg, $t_R$ = 39.0 min), **6** (1.8 mg, $t_R$ = 45.0 min), and **5** (2.7 mg, $t_R$ = 48.0 min).

**Isolation of compounds 8 and 9 from Xylaria sp. X187.** A culture was prepared by inoculating 50 mL PDB with 2 agar cubes of a 2-week old *Xylaria* sp. X187 culture. The culture was incubated for 2 weeks at 30 °C and 150 rpm. Afterwards, the pre-culture was used to inoculate 20 PDA plates (150 × 20 mm), which were kept at 26 °C for one month in the dark. Agar plates were then cut into pieces, soaked in EtOAc overnight. The crude extract (720 mg) was loaded onto C18 SPE cartridge and fractionated using a gradient solvent system of $H_2O$-$CH_3CN$ (100:0–

0:100, v/v) to give seven fractions (A-F). Xylasporin I (**8**) (1.2 mg, $t_R$ = 16.4 min) was isolated from fraction B $H_2O$-$CH_3CN$ (80:20, v/v) by semi-preparative reversed-phase HPLC (Phenomenex Luna C18, 250 × 10.0 mm i.d., 5 μm) with $CH_3CN$-$H_2O$ (1:9–3:7, v/v, gradient system, flow rate: 3 mL/min). The fraction D was subjected onto a semi-preparative reversed-phase HPLC (Phenomenex Luna C18, 250 × 10.0 mm i.d., 5 μm) and separated by $CH_3CN$-$H_2O$ (2:8–5:5, v/v, gradient system, flow rate: 5 mL/min) to yield xylacremolide B (**9**) (5 mg, $t_R$ = 27.9 min).

**Physical chemical data**

Xylasporin A (**1**): Pale yellow oil; [α]$^{25}_D$ 15.1 (*c* 0.09, MeOH); IR (KBr) ν$_{max}$ 3379, 2921, 1786, 1652, 1048 cm$^{-1}$; ECD (MeOH) λ$_{max}$ (△ε) 228 (1.7), 265 (4.1) nm; $^1$H (800 MHz) and $^{13}$C NMR (200 MHz), see Tables S1 and S2, respectively; positive HR-ESI-MS *m/z* 709.3430 [2 M + H]$^+$ (Calcd. for $C_{36}H_{53}O_{14}$, 709.3435).

Xylasporin B (**2**): Pale yellow oil; [α]$^{25}_D$ -7.4 (*c* 0.05, MeOH); IR (KBr) ν$_{max}$ 3422, 2919, 1784, 1552, 1248 cm$^{-1}$; ECD (MeOH) λ$_{max}$ (△ε) 245 (3.7) nm; $^1$H (800 MHz) and $^{13}$C NMR (200 MHz), see Tables S1 and S2, respectively; positive HR-ESI-MS *m/z* 293.1742 [M-$H_2O$ + H]$^+$ (Calcd. for $C_{17}H_{25}O_4$, 293.1753).

Xylasporin C (**3**): Pale yellow oil; [α]$^{25}_D$ -11.2 (*c* 0.08, MeOH); IR (KBr) ν$_{max}$ 3412, 2903, 1779, 1498, 1228 cm$^{-1}$; ECD (MeOH) λ$_{max}$ (△ε) 242 (3.3) nm; $^1$H (800 MHz) and $^{13}$C NMR (200 MHz), see Tables S1 and S2, respectively; positive HR-ESI-MS *m/z* 621.3627 [2 M + H]$^+$ (Calcd. for $C_{34}H_{53}O_{10}$, 621.3639).

Xylasporin D (**4**): Pale yellow oil; [α]$^{25}_D$ 8.7 (*c* 0.06, MeOH); IR (KBr) ν$_{max}$ 3405, 2895, 1770, 1454, 1213 cm$^{-1}$; ECD (MeOH) λ$_{max}$ (△ε) 251 (3.2) nm; $^1$H (800 MHz) and $^{13}$C NMR (200 MHz), see Tables S1 and S2, respectively; positive HR-ESI-MS *m/z* 351.1784 [M+Na]$^+$ (Calcd. for $C_{17}H_{28}O_6Na$, 351.1784).

Xylasporin E (**5**): Pale yellow oil; [α]$^{25}_D$ 5.3 (*c* 0.04, MeOH); IR (KBr) ν$_{max}$ 3412, 2903, 1787, 1464, 1223 cm$^{-1}$; ECD (MeOH) λ$_{max}$ (△ε) 245 (3.6) nm; $^1$H (800 MHz) and $^{13}$C NMR (200 MHz), see Tables S1 and S2, respectively; positive HR-ESI-MS *m/z* 657.3850 [2 M + H]$^+$ (Calcd. for $C_{34}H_{57}O_{12}$, 657.3850).

Xylasporin F (**6**): Pale yellow oil; [α]$^{25}_D$ 8.1 (*c* 0.05, MeOH); IR (KBr) ν$_{max}$ 3402, 2895, 1780, 1458, 1215 cm$^{-1}$; ECD (MeOH) λ$_{max}$ (△ε) 245 (3.4) nm; $^1$H (800 MHz) and $^{13}$C NMR (200 MHz), see Tables S1 and S2, respectively; positive HR-ESI-MS *m/z* 657.3855 [2 M + H]$^+$ (Calcd. for $C_{34}H_{57}O_{12}$, 657.3850).

Xylasporin I (**8**): White powder; $^1$H (700 MHz) and $^{13}$C NMR (150 MHz), see Table S20; positive HR-ESI-MS *m/z* 309.1694 [M + H]$^+$ (Calcd. for $C_{17}H_{25}O_5$, 309.1697).

Xylacremolide B (**9**): White crystals; $^1$H (700 MHz) and $^{13}$C NMR (150 MHz), see Table S23; positive HR-ESI-MS *m/z* 417.2369 [M + H]$^+$ (Calcd. for $C_{23}H_{33}N_2O_5$, 417.2384).

**Conformational search and geometry optimization.** All conformers were acquired through Marvin module (ChemAxon). Minimize molecular conformations were obtained using an industry standard MMFF94 force field and a generic Dreiding force field. Conformers proposed in this study within 5 kJ/mol found in the MMFF force field were selected for geometry optimization by Gaussian 16 program package39 at B3LYP/6-31 + G(d,p) level. The solvent (MeOH or CDCl$_3$) effect was considered in the PCM approximation (Supplementary Data 1).

**Chemical shifts calculation and DP4+ analysis.** Geometrically optimized conformers for possible diastereomers of **1–6** and **8** were proceeded to calculation of gauge-invariant atomic orbital (GIAO) magnetic shielding tensors at the B3-LYP/6-31 + G(d,p) level using Gaussian 16 program package$^{39}$. Chemical shift values were calculated from the magnetic shielding tensors using **equation 1** where δ is the calculated NMR chemical shift for nucleus *x*, and σ$^o$ is the shielding tensor for the proton and carbon nuclei in tetramethylsilane calculated with the B3-

LYP/6-31 + G(d,p) basis set (Supplementary Data 1).

$$\delta^x_{calc} = \sigma^o - \sigma^x \qquad (1)$$

The calculated unscaled NMR properties of the optimized structures were averaged based on the Boltzmann populations of conformers and scaled chemical shifts values were obtained using **equation 2**.

$$\delta_{scaled} = \frac{\delta_{unscaled} - intercept}{slope} \qquad (2)$$

Each experimental set of data was separately compared in detail with calculated ones, specifically for each accounted atom. Experimental and calculated chemical shits ($\delta$) were compared using the $\Delta\delta$ parameter according **equation 3**.

$$\Delta\delta = |\delta_{calc} - \delta_{exp}| \qquad (3)$$

where $\delta_{calc}$ and $\delta_{exp}$ are the calculated and experimental chemical shift values, respectively.

After calculating all $\Delta\delta$ values, mean absolute error (MAE) values and DP4+ probabilities were computed for determining which calculated set of data fits better with the experimental one. The MAE is defined as the summation ($\sum$) of the n computed absolute $\delta$ error values ($\Delta\delta$) normalized to the number of $\Delta\delta$ errors considered (n) (**equation 4**).

$$MAE = \frac{\sum(\Delta\delta)}{n} \qquad (4)$$

DP4+ probability analysis was processed upon using an Excel sheet from ref. 19.

**ECD spectra calculation.** ECD calculations of **1** conformers were performed at the identical theory level and basis sets with Gaussian 16 program package[39]. Calculated ECD spectra were simulated by overlying each transition, where $\sigma$ is the width of the band at the height $1/e$ (**equation 5**). $E_i$ and $R_i$ are the excitation energies and rotatory strengths for transition $i$, respectively. In the present study, the value of $\sigma$ was 0.30 eV. The excitation energies and rotational strengths for ECD spectra were calculated, and ECD visualization was performed using GaussView 6.0.

$$\triangle\varepsilon(E) = \sum_{i=1}^{n} \varepsilon_i(E) = \sum_{i=1}^{n}\left(\frac{R_i E_i}{2.29\ x\ 10^{-39}\sqrt{\pi\sigma}}\exp\left[-\left(\frac{E - E_i}{\sigma}\right)^2\right]\right) \qquad (5)$$

**Antimicrobial activity assays. A**. Antibacterial aactivity assay was performed according to the NCCLS (National Committee for Clinical Laboratory Standards) with compound (1.0 mg/mL in DMSO) and ciprofloxacin (5 μg/mL in aqua dest. (cip.) as control. Bacterial test strains included *Bacillus subtilis* sp. 6633, *Staphylococcus aureus* SG511, *E.coli* sp. SG458, *Pseudomonas aeruginosa* sp. SG137, MRSA *Staphylococcus aureus* sp. 134/94, VRSA *Enterococcus faecalis*, *Mycobacterium* vaccae sp. 10670. The antifungal activity assays were performed according to ref. 40. **B)** *Aspergillus nidulans* RMS011 was chosen as a test strain for antibacterial testing. As a reference the antifungal agent carboxine (AppliChem GmbH, Germany) was used. Stock solutions (1 mg/mL in MeOH) of the tested substances were added in 96 well plates and MeOH was evaporated at room temperature. For inoculation 150 μL of conidia suspension in aspergillus minimal medium (AMM, supplemented with 3 mg/L *p*-aminobenzoic acid and 5 mM L-arginine) and resazurin (0.002% (w/v); Sigma-Aldrich Chemie GmbH, Germany) were used. Conidia concentration was $2.5 \times 10^5$ conidia/mL. Assays were incubated at 37 °C for 20 h in a Synergy H1 plate reader (BioTek Instruments GmbH, Germany) and fluorescence was determined at the wavelengths 570 nm (excitation) and 615 nm (emission). The growth inhibition was calculated by normalization to the positive control without supplementation of substances and the negative control without inoculum. All assays were carried out in triplicates. Shown is the relative inhibition (in %) of epoxycytochalasine Q (**Ecc**), pseudoxylallemycin A (**Pam**), cytosporins N-R (**2-6**), 1-hydroxy-2-hydroxymethyl-3-pent-1-enylbenzene (**7**) and SPE fraction eluted with MeOH of *Xylaria* sp. X802 raw extract (**SPE**) on *A. nidulans* RMS011 growth after 20 h. The percentage of inhibition equal to 0 is considered normal growth and equal to 100 is complete growth inhibition. Growth control (**Control**) was set up without supplementation of compounds. The antifungal agent carboxine (**Cbx**) was used as a reference. All of the experiments were performed in triplicate and the standard deviations are shown (Supplementary Fig. S62).

**Reporting summary**
Further information on research design is available in the Nature Portfolio Reporting Summary linked to this article.

## Data availability
Supplementary Information: contains copies of GNPS figures, all NMR spectra, HRMS spectra of clean compounds, comparison of calculated and measured NMR data, DP4+ probability analyses, BLAST results of antimicrobial activity screening, and in silico analysis of the *px* gene cluster within the genome of *Xylaria* sp., and crystal data and structure refinement data for compound **9**. Supplementary Data 1: contains lists of calculated ¹H and ¹³C chemical shifts for each determined conformer. Supplementary Data 2: Crystallographic Information File (CIF) for structure of compound **9**. The X-ray crystallographic coordinates for structure of compound **9** reported in this study have also been deposited at the Cambridge Crystallographic Data Centre (CCDC), under deposition numbers CCDC-2347670. These data can be obtained free of charge from The Cambridge Crystallographic Data Centre via www.ccdc.cam.ac.uk/data_request/cif. All data generated or analyzed during this study are included in this published article and its supplementary files. NMR data files and supplementary mass-spectrometry data files have also been deposited at Zenodo under the https://doi.org/10.5281/zenodo.10978114.

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

## Acknowledgements

This work was supported by the National Research Foundation of Korea (NRF) grant funded by the Korea government (MSIT) (2019R1A5A2027340 and 2021R1A2C2007937) and National Research Foundation of Korea (NRF) funded by the Ministry of Education (2020R1A6A3A03037782). This work was also supported under the framework of international cooperation program managed by the National Research Foundation of Korea (2020K2A9A2A06037042). This study was funded by the German Research Foundation (DFG, Deutsche Forschungsgemeinschaft) under Project-ID 239748522 – CRC 1127 (project A6), project BE 4799/4-1 under Project-ID 438841444, and the Germany's Excellence Strategy under Project-ID 390713860 - EXC 2051. We thank Mrs. Heike Heinecke (Hans Knöll Institute) for recording NMR spectra and Mrs. Andrea Perner (Hans Knöll Institute) for HRMS measurements. We thank Dr. Fabien Fontaine-Vive for launching DFT and TDDFT calculations as well as Université Côte d'Azur and OPAL infrastructure for providing Azzurra High-Performance Computing. Instrumentation and technical assistance for crystallographic data were provided by the Service Center X-ray Diffraction, with financial support from Saarland University and German Science Foundation project number INST 256/506-1 (D8 Venture).

## Author contributions

S.R.L., M.D., J.F., H.G., F.S., S.B., J.S.Y., B.S., C.B. designed research; S.R.L., M.D., J.F., H.G., S.B., F.S., S.B., J.S.Y., performed the experiments; S.R.L., M.D., J.F., H.G., F.S., J.S.Y., B.M., S.Y.J., C.B., K.H.K. analysed data; B.S., K.H.K., C.B. supervised, provided funds and infrastructure. S.R.L., M.D., J.F., H.G., F.S., B.M., J.S.Y., C.B. generated the figures and S.R.L., M.D., J.F., B.S., K.H.K., C.B. wrote the manuscript with input from all authors

## Funding

## Competing interests

The authors declare no competing interests. H.G. is a Senior Editor for Communications Chemistry, but was not involved in the editorial review of, or the decision to publish this article.
