## [Peer review file · Communications Chemistry]

Molecular networking and computational NMR analyses uncover six polyketide-terpene hybrids from termite-associated *Xylaria* isolatesReviewers' comments:

Reviewer #1 (Remarks to the Author):

This study reports a series of putative terpene-polyketide hybrids from a fungus of the genus *Xylaria*, which is here referred to under the subgenus name "Pseudoxylaria".

The editors and authors should be advised that strictly speaking, this is not correct according to the standing rules of taxonomy! If a genus is subdivided into subgenera and sections, the species placed in these subgeneric taxa retain the genus name.

I also tried to find further information on the "strain X802" in reference 7 but failed to do so. That paper (Visser et al 2009) does not apparently give any information as to where the strains have been deposited and a strain 802 is not listed. Ref. 7 should actually not have been published in such a renowned journal, since it normally regarded absolutely important to deposit vouchers as a prerequisite for publication there. If the strain is indeed derived from the old study that was published ca. 15 years ago, the authors should clarify the origin and also explicitly state where this fungus is deposited. It could well be that it is extant in the FABI collection as it was originally obtained from South Africa and scientists from FABI are co-authors of the respective publications, or as the first author is Dutch it could have ended up with CBS/WFBI?.

This said, I am already at the end of my criticism because the way in which the new compounds were isolated, identified and characterized is state of the art. The structure elucidation seems very sound, perhaps except for the ECD calculations to somewhat tentatively determine the absolute configuration, which can never replace a total synthesis or an X-ray structure. There are already some examples showing that this ECD calc. stuff does not always work out. On the other hand, it seems that these calculations have become a kind of new fashion and other reviewers may actually insist that such data be included. There are even IR data included, which shows that the authors were very careful to provide as much information on their new compounds as possible.

The MS-MS based dereplication in combination with 2D-MMR data and DP4+ probability analyses is a rather interesting and innovative approach. The theory about the biosynthetic origin deduced from transcriptomics and genome analyses is also very sound, even though purists may say that this methodology can ultimately not replace classical feeding experiments.

The cited literature is almost complete and very comprehensive regarding the cytosporins and other related metabolites. The discussion and the conclusion are very sound. The whole ms is rather well written aside from some typos and the spectral data in the SI seem to support the structure proposals very well.

I have made some small annotations in the pdf file and recommend the paper for acceptance after minor revisions.

Reviewer #2 (Remarks to the Author):

Lee et al. describe the discovery and structure elucidation of six PKS-terpene hybrids from *Pseudoxylaria*. Compounds were discovered efficiently using molecular network analysis based on MS/MS data. Structure elucidation was done using NMR and MS experiments, comparison with previous spectral data, and computational chemistry. Furthermore, the biosynthetic pathway was deduced from the analysis of putative biosynthetic gene loci found in the genomic data. Although the novelty of the structures and biosynthesis of the compounds is somewhat lacking in impact, the methods employed and the analytical data are reliable from a chemical point of view, and there appears to be a certain value in publishing them as a paper. However, the following points should be improved or revised.

1. In Abstract. I am not sure about the last sentence in the abstract. First of all, I don't think the expression is grammatically very favorable. Second, is the analysis/discussion you have done related to the biosynthetic origin of xylasporins?
2. In Figure 1. As can be said throughout this paper, there is no correspondence between each image and the text, such as A-E in the main figure. On the other hand, there is a detailed correspondence with SI. Figure 1 is somewhat better, but the subsequent figures are completely unable to guess the correspondence with the text. These points require significant improvement. Going back to Figure 1, I have no idea why Fig. 1B and Fig. 1C need to appear here. I can guess that Fig1E is related to the culture method, but please mention it in more detail in the text.
3. In Figure 3. The carbon No. in the structural formula written in Fig. 3C is off. Please correct the main text to correspond to Fig. 3A-C. I am not sure what the dashed squares mean. Please map it to the main text.
4. In Figure 4. Please add that the arrows indicate NOESY correlations in the figure of the relative three-dimensional arrangement in Fig. 4A. Please map the main text properly to Fig 4.
5. In Figure 5. Please respond in the same way as Fig. 4.
6. In Figure 6. The possible biosynthetic pathway is proposed in the second half of Fig. 6, explained in paragraph P8, but it is too brief and it is difficult to understand how each step is done. Please try to explain in more detail.

Reviewer #3 (Remarks to the Author):

The manuscript by Kim, Beemelmans, and coworkers describes the discovery of six new fungal compounds, xylasporins A-F, from *Pseudoxylaria* sp. X802 by application of MS/MS molecular networking

and cocultivation with *Termitomyces* sp. T153 enabling comparative metabolomics. The compounds were isolated from fungal cultures grown on PDA plates and structurally characterized by in-depth NMR spectroscopic, ECD, and computational analysis, leading to convincing assignments of their complete stereostructures. Bioinformatic comparison of the putative xylasporin biosynthetic pathway to known pathways encoding structurally related (precursor) metabolites led to a putative theoretical biosynthetic assembly. Preliminary biological activity studies, exclusively against *A. nidulans*, did not show any significant antifungal activity. Overall, this is very solid work, in particular concerning the structure elucidation. The paper is well written and illustrated. However, the identified compounds are structurally highly related to the long-known cytosporins, essentially only diverging in the length of the C8 side-chain (being shorter in the new compound series). No new biosynthetic insights or exciting biological activities are reported. The degree of novelty of the findings might thus be considered relatively low. I still believe that the findings are interesting to natural product discovery groups, particularly also to chemical biologists. Acceptance of the work in *Communications Chemistry* still seems borderline in its current state. If the authors could provide more biological data (broader testing against fungi – also with relevance to the natural environment of the producer strain; further profiling against bacteria / cytotoxicity evaluation), the significance of the work could be considerably improved. Currently, publication in *Sci. Rep.* or a specialized natural product chemistry journal (*JNP*) would seem more appropriate.

Minor comments: there are some minor typos / missing words throughout the manuscript. In addition, the subchapter “antibacterial activity studies” seems to only contain a single antifungal test system -> please adjust.

Molecular networking analysis and computational NMR analysis uncovers six polyketide-terpene hybrids of the stowaway fungus *Pseudoxylaria*

Seoung Rak Lee,^{#[a][b]} Marie Dayras,^{#[c]} Janis Fricke,^[d] Huijuan Guo,^[d] Felix Schalk,^[d] Jae Sik Yu,^{[a][e]} Se Yun Jeong,^[a] Christine Beemelmans,^{*[c][d][f]} and Ki Hyun Kim^{*[a]}

- [a] School of Pharmacy
Sungkyunkwan University
Suwon 16419, Republic of Korea
- [b] College of Pharmacy and Research Institute for Drug Development
Pusan National University
Busan 46241, Republic of Korea
- [c] Anti-infectives from Microbiota
Helmholtz Institute for Pharmaceutical Research Saarland (HIPS)
Campus E8.1, 66123 Saarbrücken, Germany
- [d] Chemical Biology of Microbe-Host Interactions
Leibniz institute for Natural Product Research and Infection Biology – Hans-Knöll-Institute (HKI)
Beutenbergstraße 11a, 07745 Jena, Germany
- [e] Department of Integrative Biological Sciences and Industry
Sejong University
Seoul 05006, Republic of Korea
- [f] Saarland University, 66123 Saarbrücken, Germany

* Corresponding authors:

equal contribution

Ki Hyun Kim, Tel: +82-31-290-7700; Fax: +82-31-290-7730; E-mail: khkim83@skku.edu

Christine Beemelmans, E-mail: christine.beemelmans@helmholtz-hips.de

Abstract: Fungi constitute the Earth's second most diverse kingdom, however only a small percentage of these have been thoroughly examined and categorized for their secondary metabolites, which still limits our understanding of the ecological chemical and pharmacological characteristics of fungi. In this study, we explored members of the co-evolved subgenus *Pseudoxylaria* and identified a family of highly oxygenated chromene derivatives from culture extracts of *Pseudoxylaria* sp. X802 using an MS/MS molecular networking-based dereplication approach. Extensive comparative spectrometric (HRMS²) and spectroscopic (1D and 2D NMR) studies allowed to characterize six yet unreported xylasporins A-F (**1-6**), of which xylasporin A (**1**) featured a rare cyclic-carbonate moiety. Configurations of stereogenic centers were deduced from comparative ROESY and chemical shift analysis across the compound family, which was supported by DP4+ probability analyses. The absolute configurations of compound **1** was proposed based on TDDFT-ECD calculations. The putative biosynthetic origin was deduced by comparative genomic and transcriptomic analysis, which will allow in future for the engineering of the biosynthetic pathway, which was found to be **wide spread** across the fungal kingdom and of diverse nature.

Introduction

The fungal kingdom ranks as the second most diverse kingdom on Earth, estimated to comprise 3-4 million species.¹ Within this kingdom, Sordariomycetes stands out as a diverse and significant class, including fungal families such as *Hypoxylaceae* and *Xylariaceae*,^{1,2} which have recently gained considerable attention due to their enormous biosynthetic repertoire to produce secondary metabolites with diverse biological activities.^{3,4} While the Xylariaceae family includes predominately free-living saprophytic members that are often noticeable due to the development of stromatal tissue on decaying wood, it also includes the distinctive subgeneric taxon *Pseudoxyllaria*, whose members have exclusively reported from fungal garden material of fungus-farming termites, in which termites cultivate the fungal mutualist *Termitomyces* as a crop.^{5,6} *Pseudoxyllaria* fungi exclusively appear in the fungus comb of termite colonies that are weakened or abandoned, rapidly dominating any remaining termite food source (Figure 1).^{7,8,9} In our recent investigation, we delved into the genomic and biochemical foundations of this distinctive co-evolved antagonistic behavior and substrate specialization,¹⁰ and posited that the secreted antimicrobials of *Pseudoxyllaria* could play a role in outcompeting the fungal mutualist once *Pseudoxyllaria* emerges.^{11,12,13,14}

First transcriptomic and metabolomic studies indeed showed that metabolite production is triggered and diversified when *Pseudoxyllaria* species were cultivated on comb-mimicking growth conditions and in fungus-fungus co-cultivations against the fungal cultivar.^{10,15} To further solidify the hypothesis, we re-investigated the secondary metabolome of the model strain *Pseudoxyllaria* sp. X802^{7,11} using eco-mimetic cultivation conditions and a comparative HRMS²-based metabolomic approach, which led to the isolation and characterization of six yet unreported polyketide-terpene hybrid metabolites named xylasporins A-F (1-6). Due to the structural complexity of this compound class, we pursued chemical-computational investigations and compared the relative and absolute structures reported from this compound class. Our comparative genomic studies also allowed deducing the putative polyketide-terpene-based biosynthetic origin from which the formation of multiple, yet unreported, cytosporin derivatives could be deduced.

Figure 1. Fungus comb overgrown by *Pseudoxyllaria* strains seven days after incubation in the absence of termites.

Results and Discussion

Cultivation and comparative analysis

Pseudoxyllaria sp. X802⁷ was cultivated in co-culture with the fungal mutualist *Termitomyces* sp. T153 (7-14 days, 30 °C) (Figure 2A). Culture extracts were analyzed for (induced) metabolite production by comparing the metabolome of the interaction zone (ZOI) to metabolites secreted from an axenic agar culture grown on potato-dextrose agar (PDA) using liquid chromatography (LC) coupled with tandem electron-spray ionization (ESI)-HRMS/MS analyses. Obtained MS²-data was dereplicated using the Global Natural Product Social Molecular Networking Web platform (GNPS) (Figure S1).¹⁶ The comparative molecular network revealed several molecular ion clusters containing more than ten different nodes, including a cluster putatively assigned to diketopiperazines, a cluster assigned to previously identified pseudoxyllallemycins (m/z 549.322, 617.456, 619.262, and 685.394)¹¹ and a cluster including cytochalasin derivative (m/z 524.263) (Figure S2).¹⁰ One additional MS²-cluster was found, which included several spectral nodes with yet unknown molecular ion features, and while representatives of each cluster were found within all samples, a few nodes were only detectable in samples derived from the interaction zone of a co-culture experiment (Figure 2D). We then investigated the abundances of these features within culture extracts obtained from cultures grown on rice-sawdust medium or on glass beads (Figure 2A-C),¹⁰ which provided structural elements for fungal growth. Again, almost all features were detectable, but the intensity of each molecular ion features was depended on the substrate provided for growth, which indicated a subtle transcriptional regulation of tailoring enzymatic activity.¹⁰

Figure 2. A) Co-cultures of *Pseudoxyllaria* sp. X802 versus *Termitomyces* sp. T153 on PDA/soil medium; B) culture of *Pseudoxyllaria* sp. X802 on glass beads with liquid PDB, and C) on rice-sawdust medium. D) Expanded GNPS-based molecular network cluster of xylasporin-based molecular ion features. Features that were only present in the zone of inhibition (ZOI) are of pink color and those from axenic growth of X802 are colored in blue (1, m/z 337.164 [M-H₂O+H]⁺, 2/3, m/z 293.174 [M-H₂O+H]⁺, and 4/5/6 m/z 311.184 [M-H₂O+H]⁺). E) Selected ion chromatogram of 1 (m/z = 337.1636 [M-H₂O+H]⁺), 2/3 (m/z = 293.1740 [M-H₂O+H]⁺), 4/5/6 (m/z = 311.1847 [M-H₂O+H]⁺) and 7 (m/z = 193.1223 [M+H]⁺) within extracts derived from agar plate cultivation (ii), glass beads cultivation with liquid PDB (iv) and rice-sawdust medium (vi) of three weeks old *Pseudoxyllaria* sp. X802. The corresponding negative controls without X802 are shown below each lane (i, iii and v), respectively.

Structural elucidation of xylasporin derivatives from X802

We then pursued the isolation of the detected unknown features. For this, the fungus X802 was grown for 14 days on PDA plates and mycelium-covered plates were then extracted with methanol to yield an enriched crude extract. Compounds were purified by MS- and UV-guided reverse-phase chromatography. Overall, we were able to isolate seven metabolites, including a known compound, named 1-hydroxy-2-hydroxymethyl-3-pent-1-enylbenzene (**7**).¹⁷ Comparative NMR analysis of compounds **1-6** uncovered a similar 1D and 2D NMR pattern, which indicated towards a shared bicyclic, and oxidized core skeleton.

The molecular formula of compound **1**, C₁₈H₂₆O₇, was deduced from HRESI(+)-MS data [*m/z* 709.3430 [2M+H]⁺, calcd. for C₃₆H₅₃O₁₄, 709.3435]. The ¹H NMR data of **1** showed the presence of signals for three methyl groups, three methylene units, one oxygenated methylene, four oxygenated methines, and two olefinic protons (Tables S1 and S2). HSQC and HMBC spectra revealed a total of 18 carbon signals attributable to three methyls, four methylenes, six methines, four quaternary carbons, and one carbonyl-like carbon (Table S1). Interpretation of 2D NMR spectra allowed the determination of planar structure of **1**, with notable HMBC correlations between H-7/C-14, and H-6/C-19 and H-7/C-19 showing the presence of a cyclic carbonate group, as well as COSY correlations of an aliphatic spin system. Overall compound **1** showed similarities to cytosporin E featuring a pentenyl chain at C-8 compared to a heptenyl chain in cytosporin E.¹⁸ The relative configuration of **1** was determined by vicinal ¹H-¹H coupling constants (³J_{HH}) and ROESY correlations of H-3/H₃-12, H-10/H₃-11, H₃-11/H-4 α , and H-6/H-4 β indicating the equatorial orientation of the hydroxyl group at C-3 and the axial orientation of H-6, H-7, and H-10 (Figure 3). As the hexahydrobenzopyrane moiety of compound **1** showed NMR chemical shifts very close to those of cytosporin E, we deduced that these compounds share the same absolute configuration: 3*S*, 5*R*, 6*R*, 7*R*, and 10*S* (Table S3).

Figure 3. A) Planar structure of xylasporin A (**1**) and key 2D NMR correlations for compound **1**; B) Experimental ECD spectrum for compound **1** and Boltzmann-averaged TD-DFT calculated ECD spectra for **1a** (3*S*, 5*R*, 6*R*, 7*R*, 10*S*), **1b** (3*R*, 5*S*, 6*S*, 7*S*, 10*R*); C) Proposed absolute structure of xylasporin A (**1**) and reported carbonate containing natural products of fungal origin.

To support the assumption, we pursued Gauge-including atomic orbital (GIAO) NMR chemical shifts calculations followed by DP4+ probability analysis,¹⁹ which confirmed the *cis*-orientation of the angular substituents at C-5 and C-10 (Tables S4-S5, Figure S9). Additionally, we compared the experimental ECD spectrum of **1**, which exhibits two positive Cotton effects at 228 nm and 265 nm, with the calculated ECD spectra for the most stable conformers of the two possible enantiomers of **1** [**1a** (3*S*, 5*R*, 6*R*, 7*R*, 10*S*) and **1b** (3*R*, 5*S*, 6*R*, 7*S*, 10*R*)], which in summary allowed us to deduce the absolute configuration for compound **1** (3*S*, 5*R*, 6*R*, 7*R*, 10*S*) (Figure 3). Thus, we named compound **1** xylasporin A due to its structural resemblance to the cytosporin family.

20,21,22,23,24,25,26,27,28,29 Here, it was intriguing to note that cyclic-carbonate patterns are rarely found in natural products, and have so

far predominately been reported from the marine-derived fungus *Eutypella scoparia*,²⁰ *Aspergillus* sp. PSU-RSPG185 (aspergillusols),²³ and *Phoma* sp. (phomoxins).³⁰

The molecular formula of xylasporins **2** and **3** was determined as C₁₇H₂₆O₅ based on the HRESI(+)MS analyses, which showed pseudo-molecular ions peaks at *m/z* 293.1742 [M-H₂O+H]⁺ (calcd. for C₁₇H₂₅O₄, 293.1753) for **2** and *m/z* 621.3627 [2M+H]⁺ (calcd. for C₃₄H₅₃O₁₀, 621.3639) for **3**, indicating that these compounds are likely structural isomers. The 1D and 2D NMR spectra of **2** and **3** exhibited some similarities with those signals assigned to the core structure of **1**. However, signals for the carbamate group were lacking and chemical shift patterns at position C-5, C-6 and C-7 were suggestive for the presence of an epoxide group at C-5 and C-6 (Tables S1 and S2). Comparison with literature data suggested that compounds **2** and **3** had a similar planar to cytosporins D and M, but carried a pentenyl chain at C-8 in contrast to a heptenyl chain in cytosporins D and M.^{18,27} Differences in chemical shifts and ³J_{HH} coupling constants for H₂-4, as well as key ROESY correlations supported the notion that compounds **2** and **3** share the same relative configurations 3*S*^{*}, 5*R*^{*} and 6*S*^{*} with an inversion of relative configurations for C-7 and C-10 (**2**: 7*R*^{*}, 10*S*^{*}; **3**: 7*S*^{*}, 10*R*^{*}) (Figure 4). Comparison of ¹H and ¹³C NMR chemical shifts of the hexahydrobenzopyrane moieties of compound **2** and cytosporin D led us to deduce the absolute configuration of **2** as 3*S*, 5*R*, 6*S*, 7*R*, and 10*S* [$\overline{X}(|\Delta\delta_{\text{H}(2-\text{cystoporin D})}|) = 0.04 \text{ ppm}$, $s = \Delta\delta_{\text{H}(2-\text{cystoporin D})} = 0.05 \text{ ppm}$; $\overline{X}(|\Delta\delta_{\text{C}(2-\text{cystoporin D})}|) = 0.24 \text{ ppm}$, $s = \Delta\delta_{\text{C}(2-\text{cystoporin D})} = 0.15 \text{ ppm}$] and for compound xylasporin C (**3**) as 3*S*, 5*R*, 6*S*, 7*S*, and 10*R* (Tables S6-S9).¹⁸ These deductions were solidified by GIAO NMR chemical shifts calculation followed by DP4+ probability (Tables S10-S13, Figures S22-S23).

Figure 4. A) Proposed structure of xylasporin B (**2**), structure with $|\Delta\delta_{\text{C}(2-\text{cystoporin D})}|$ values in ppm and 3D depiction showing ROESY correlations; B) Proposed structure of xylasporin C (**3**), structure with $|\Delta\delta_{\text{C}(3-\text{cystoporin D})}|$ values and 3D depiction showing ROESY correlations; C) Structures of reported natural products with similar core scaffolds used for comparison.

The molecular formulas of xylasporins D (**4**), E (**5**), and F (**6**) were deduced from the sodium-adducted ion peak at *m/z* 351.1784 [M+Na]⁺ (calcd. for C₁₇H₂₈O₆Na, 351.1784) from HRESI(+)MS data of **4**, and from the pseudo-molecular ions peaks at *m/z* 657.3850 [2M+H]⁺ and *m/z* 657.3855 [2M+H]⁺ (calcd. for C₃₄H₅₇O₁₂, 657.3850) from HRESI(+)MS data of **5** and **6**.

Interpretation of the ¹H and ¹³C NMR spectra of these compounds revealed again the hexahydrobenzopyrane scaffold with chemical shift differences at C-5, C-6 and C-7 compared to **1**, which suggested the presence of hydroxyl groups at C-5, C-6, and C-7. COSY and key HMBC correlation analyses allowed deducing the planar structures of **4**, **5**, and **6**, resembling that of cytosporin L, which includes a pentenyl chain at C-8 compared to a heptenyl chain in cytosporin L (Figure 5).²⁰ Relative configurations of C-3, C-5, C-6, C-7 and C-10 of **4** were deduced from key ROESY correlations (Figure 5) and analogous GIAO NMR chemical shifts calculations followed by DP4+ probability, which all together supported the proposed configuration of 3*S*^{*}, 5*S*^{*}, 6*R*^{*}, 7*R*^{*}, and 10*R*^{*} (Tables S14-S15, Figure S30).

We again compared the ^1H and ^{13}C chemical shifts for compound **5** and **6**. Here it is noteworthy that the chemical shift values of C-6 and C-7, and the vicinal coupling constant between the protons H-6 and H-7 (**4**: 6.5 Hz vs. **5** and **6**: 4.5 Hz) in relation to those values obtained for compound **4** were suggestive for an *S*-stereochemistry at position C-6 and C-7; a notion, which was also supported by key ROESY correlations (H-6/H-10 for **5** and H-6/H-4 β for **6**). Stereochemistry at position C-10 was deduced for both compounds from signatory ^{13}C chemical shift patterns and ROESY correlations (**5**: H-10/H-3, H-3/H₃-12, and H₃-12/H-4 α ; **6**: H-10/H-4 α and H-4 α /H₃-11). The stereochemical configuration of C-5 in compound **5** remained an open question as the experimental and calculated NMR data resulting of DP4+ probability of 78% (Tables S16-S17, Figure S37). While still ambiguous, we propose that ROESY correlations between H-10 and H-3, and H-10 and H-6 are suggestive for a 5*S* configuration for compound **5**. In contrast, the stereochemistry at C-5 for compound **6**, was determined based on the key H-10/H-4 α ROESY correlation and unambiguous DP4+ analysis of 100% (Tables S18-S19, Figure S44). Overall, the combination of NMR interpretation and calculation allowed us to proposed the absolute configuration for **5** as 3*S*,5*S*,6*S*,7*S*,10*R* and for **6** as 3*S*,5*R*,6*S*,7*S*,10*S*.

Last but not least, a known biosynthetic precursor, 1-hydroxy-2-hydroxymethyl-3-pent-1-enylbenzene (**7**) was isolated and characterized according to previous data (Figure 5).¹⁷

To solidify our structural assumptions, we also calculated the ^1H and ^{13}C shifts using optimized settings for Gauge-including atomic orbital (GIAO) NMR chemical shifts calculations, which was followed by DP4+ probability analysis of several stereoisomers of the three structure types that encompass compound **1-6**. Based on a general analysis, we were able to deduce patterns ("rule of thumbs") that correlated to a specific stereochemical assignments. Through a comprehensive analysis, we identified patterns or "rule of thumbs" that correlated with specific stereochemical assignments. Overall, we were able to deduce a correlation between the stereochemistry of C-3 and C-10 and the ^{13}C chemical shifts of C-11 and C-12. In case the same stereochemistry (*S*, *S* or *R*, *R*) was observed for C-3 and C-10, the ^{13}C chemical shifts of C-11 and C-12 differed by more than 10 ppm. In case the stereochemistry of C-3 and C-10 were inverted (*R*, *S* or *S*, *R*), the ^{13}C chemical shift values for C-11 and C-12 differed less than 3 ppm.

Figure 5. Proposed structure of xylasporins D (**4**), E (**5**), and C (**6**) and compound **7**. Key 2D NMR correlations found for compounds **4**, **5** and **6**.

Biosynthetic considerations

Cytosporins and xylasporins share the same planar hexahydrobenzopyran backbone, with xylasporins carrying a pentenyl chain at C-8 compared to a heptenyl chain in cytosporins. The shared planar hexahydrobenzopyran backbone suggested a mixed polyketide and terpene-type biosynthetic origin.^{17-30,31} Thus, we revisited our previous comparative genome analysis of *Pseudoxyllaria* genomes,¹⁰ in which we uncovered that four strains (X802, Mn132, Mn153, and X187) contained a cluster region bearing resemblance to the *fog* BGC (*Aspergillus ruber*)²² encoding the biosynthesis of glaucin-type of scaffolds with 50-98% amino acid identity, based on which the two congeners xylasporins G and I were isolated from strain X187.¹⁰ A detailed analysis of the *fog*-like cluster arrangements within the four *Pseudoxyllaria* genomes uncovered variations in the abundance and arrangement of several accessory genes coding for a cupin protein (pxF), a short chain oxidoreductase (pxB; SDR), and an additional Snoal-like polyketide cyclase (pxP), which likely explains the structural variations observed between xylasporins retrieved from different strains.¹⁰ We assigned the locus within the genome of X802 as the *px* gene cluster (BGC) region (Figure 6, Table S20), which encompasses a highly-reducing PKS (HRPKS) *pxA* (KS-AT-DH-ER-KR-ACP), an ABBA type aromatic prenyltransferase *pxH*, three short-chain dehydrogenases/reductases (SDR) *pxB/pxD/pxJ*, two cupins *pxC/pxF*, a cytochrome p450 monooxygenases (CYP) *pxE*, two FAD-dependent oxidoreductases *pxG/pxI*, two fungal transcription factors *pxK/pxM*, a major-facilitator superfamily (MFS) transporter *pxL*, a phosphopantetheinyl transferase *pxN* and a thioesterase domain containing enzyme *pxO*.³²

Figure 6. A) Schematic depiction of the *px* BGC in *Pseudoxyllaria* sp. X802 and comparison to homologues BGCs from the biosynthesis of flavoglaucin (*Aspergillus ruber* CBS 135680), trichoxide (*Trichoderma virens* Gv29-8) and sordarial (*Neurospora crassa* OR74A). Shown are genes coding for highly-reducing polyketide synthases (blue; HR-PKS), short-chain dehydrogenases/reductases (red; SDR), cupins (grey), cytochrome p450 monooxygenases (green; CYP), flavin dependent oxidoreductases (yellow; OR), aromatic prenyltransferases (purple; PT), transcription factors (black) and major facilitator superfamily transporters (dark grey; MFS). Non-related genes are shown in light grey. B) Putative biosynthetic assembly line yielding isolated biosynthetic precursor 7, which is likely biotransformed through a series of oxidative transformation steps to yield epoxidized cytosporins of type 2/3, tetradials 4-6 and in a yet unidentified step carbonate 1.

A detailed *in silico* of the *px* BCG uncovered that seven of the encoded proteins (PxA-E and PxG-I) showed high similarity (42-86 % similar amino acid sequence; Table S20) to the characterized FogA-F from *Aspergillus ruber* CBS 135680, which is responsible for the biosynthesis of the prenylated salicylic acid derivative flavoglaucin.³¹ Homologues gene sequences were also previously identified in other ascomycetes like *Trichoderma virens* Gv29-8 (61-80 %)³³ and *Neurospora crassa* OR74A (47-76 %),³⁴ encoding for the biosynthesis of the prenylated PKS-based salicylic acid derivatives sordarial and trichoxide, respectively. Based on these homologies, an iterative incorporation of five malonyl-CoA onto an acetyl-CoA starter unit by the HRPKS PxA was proposed to yield ACP-bound intermediate I. It was intriguing to note that xylasporins in general carried a pentenyl chain at C-8 compared to a heptenyl chain in cytosporins, which suggested yet unknown structural and functional difference in the iterative mode of the HRPKS PxA. The partial reduction of the polyketide chain should then be realized by the KR, DH and ER domains of PxA as well as PxB and PxC, similar to mechanisms that have been shown for flavoglaucin biosynthesis in *A. ruber*.³¹ Release of the polyketide II is proposed to be catalyzed by SDR PxD or the putative *trans*-acting TE domain containing protein PxO leading to the production of the putative precursor 7. Hydroxylation at C-3 by PxE and prenylation at C-5 by PxH could lead to the synthesis of a flavoglaucin congener (IV). The double bond at the prenyl group could be epoxidized by either the putative cytochrome p450 monooxygenases PxF or PxG. The chromene core structure of VI is likely formed by spontaneous ring closure leading to two distinct stereoisomers ($m/z = 275.164 [M-H_2O+H]^+$). Subsequent oxidation of the alcohol at C-7 by either Pxl or PxJ might form an unsaturated enone VIII, while a second epoxidation by PxF or PxG could catalyze the formation of 2 and 3. Hydrolysis of the oxirane ring and reduction of the carbonyl functionality should provide the three isomers 4-6, but the enzymatic basis for the regioselective introduction of the cyclic carbonate moiety still remains enigmatic.

Antibacterial activity studies

Cytosporins A-C, isolated from an endophytic strain of the taxonomically related fungus *Cytospora*,²³ were reported to act as angiotensin II binding inhibitor, while cytosporin L was shown to have activity against the bacteria *Micrococcus lysodeikticus* and *Enterobacter aerogenes* with MIC values of 3.12 μ M.¹⁸ However, cytosporin D and E were reported to have neither antibacterial nor antifungal activity against *Staphylococcus aureus*, *Escherichia coli*, and *Candida albicans*.^{23,27} As xylasporins A-F (1-6) were identified in the ZOI of the fungus-fungus co-culture, their antifungal activity against *A. nidulans* RMS011 as test strain was evaluated. However, only very mild antifungal activity in form of germination inhibition was observed for xylasporins B (2) and C (3) to a similar extent as co-isolated epoxyxylasporin Q and pseudoxyllallemycin A (Figure S48).

Conclusion

Members of the subgenus *Pseudoxyllaria* sp. are substrate-specialized and emerge from abandoned fungus comb material of fungus-growing termites. Building on the discovery that *Pseudoxyllaria* inhibits the growth of the fungal crop *Termitomyces* assisted by its biosynthetic capabilities to produce secondary metabolites, we substantiated the hypothesis by conducting experiments under ecomimetic growth conditions and employing a comparative HRMS²-based metabolomic approach. Dereplication of metabolic features, whose abundances were dependent on the provide growth conditions, led to the discovery of six new secondary metabolites of PKS-terpenoid origin. We named these cylasporins A-F (1-6) due to their hexahydrobenzopyran backbone and structural resemblance to cytosporins. GIAO NMR calculations, combined with 2D NMR interpretation were used to complete the structural elucidation of these complex molecules. The observed variation in the oxygenation pattern and the unsaturation degree within the alkenyl chain were deduced from the predicted PKS arrangement.

Supporting Information

General Experimental Procedures

Chemicals. All media, solvents and, chemicals were purchased as follows: methanol, dichloromethane, ethyl acetate, acetonitrile (Th. Geyer, Renningen); water for analytical and preparative HPLC (Millipore, Germany), formic acid (Carl Roth, Germany), MeOH-*d*₄ (Carl Roth, Germany), media ingredients (Carl Roth, Germany). Merck precoated silica gel F254 plates and RP-18 F254s plates were used for thin layer chromatography (TLC). Spots were detected on TLC under UV light or by heating after spraying with anisaldehyde-sulfuric acid.

Strains and culture conditions. A pre-culture of *Pseudoxyllaria* sp. X802 was prepared by inoculating 200 mL PDB with 1-2 agar cubes of a 2-week old *Pseudoxyllaria* sp. X802 culture and incubated for 2 weeks at 30 °C and 150 rpm. Afterwards, the pre-culture was used to inoculate 20 PDA plates (92x16 mm), petri dishes filled with glass beads containing 10 mL liquid PDB and 10 mL rice-sawdust medium mixed in a 1:1 (v/v) ratio. The cultures incubated for 3 weeks at RT. Media served as negative controls. The cultures from liquid PDB and rice-sawdust medium were extracted with each 50 mL ethyl acetate overnight. Residues were combined, evaporated under reduced pressure, dissolved in MeOH and analysed using UPLC-ESI-HRMS.

Co-cultivation studies. Co- *Pseudoxyllaria* sp. X802 (termed X802) and *Termitomyces* sp. T153 (termed T153) were cultivated on PDA plate for three weeks as inoculum.^{10,11} The mycelium of strain T153 was scratched from the surface of PDA plate and collected in sterile PBS (20 mL) to make nearly homogeneous mycelium suspension by vortex. Due to the different growth rate of both fungal strains, two different co-culture growth conditions on standard petri dish (9 cm diameter) containing 20 mL of 1/3 PDA medium were set-up. **Method A:** X802 and T153 were inoculated simultaneously: one mycelium containing agar plug of X802 (1 x 1 cm) was placed in the middle of petri dish. Then 200 μ L of T153 mycelium suspension was placed on the empty space of the same petri discs and evenly distributed by gentle spreading. **Method B:** X802 was grown for one week until vegetative mycelium of X802 was visible. Then T153 mycelium suspension was introduced close to the X802 colony. Axenic cultures of X802 and T153 were cultivated for the same time period as controls. To analyze the secreted metabolites within the zone of inhibition, the interaction zone (**Figure S1-c**) from two-week co-cultivation plate was separated, cut into small pieces and extracted by 20 mL of dichloromethane (DCM) overnight. Similarly, mycelium of X802 and T153 were separated, cut into small pieces and extracted by 20 mL of DCM for overnight respectively. Controls (X802 and T153) were extracted in the same way. The DCM extract were collected by filtration and concentrated under reduced pressure using a SpeedVac. The resultant DCM extract were dissolved into LCMS grade MeCN as 0.1 mg/mL and submitted to UPLC-ESI-HRMS analysis.

HR-MS. High resolution ultra-high performance liquid chromatography electrospray ionization mass spectrometry (UPLC-ESI-HRMS) measurements were performed on a Dionex Ultimate3000 system coupled with a Luna Omega C18 column (100 x 2.1 mm, particle size 1.6 μ m, pore diameter 100 Å, Phenomenex) combined with Q-Exactive Plus mass spectrometer (Thermo Scientific) equipped with an electrospray ion (HESI) source. Column oven was set to 40 °C; scan range of full MS was set to m/z 150 to 2,000 with resolution of 70,000 and AGC target 3e6 and maximum IT 100 ms under positive and negative mode with centroid data type. MS² was performed to choose top10 intensive ions under positive mode with resolution of 17,500 and AGC target 1e5 and maximum IT 50 ms and (N)CE (20/30/40) with centroid data type. The spray voltage (+) was set to 4000 volt, and (-) was set to 3300 volt. The capillary temperature (+/-) was set to 340 °C and probe heater temperature (+/-) was set to 200 °C. The sheath gas flow (+/-) was set to 35 L/min and Aux gas flow (+/-) to 5 L/min. Max spray current (+) and (-) was set to 100 volt. S-Lens RF level was set to 50.

GNPS analysis. Culture extracts were submitted to LC-ESI-HRMS metabolomics analysis under standard condition. Metabolites were separated under the gradient: 0 – 0.5 min, 5% B; 0.5 – 18 min, 5% – 97% B; 18 – 23 min, 97% B; 23 – 25 min, 97% – 5% B; 25 – 30 min, 5% B (A: 0.1% FA; B: MeCN with 0.1% FA), with flow rate of 0.3 mL/min and injection volume is 5 μ L. Metabolomics raw data acquired on a Thermo QExactive Plus mass spectrometer was converted to 32-bit mzXML files using MSConvert GUI (ProteoWizard), in order to generate a mass spectral molecular networking using the global natural products social molecular networking platform (GNPS, <https://gnps.ucsd.edu>).¹² Data analysis used default parameters, except for the cosine threshold, set to 0.7, minimum matched fragment ions of 4, network TopK 10, and for the tolerances of the precursor- and fragment ion masses, both set to 0.02 Da. The mass spectral network was assembled and visualized using Cytoscape.³⁵

Analytical and preparative HPLC. High-performance liquid chromatography (HPLC) were performed on a Waters 1525 Binary HPLC pump with a Waters 996 Photodiode Array Detector (Waters Corporation, Milford, CT, USA). Semi-preparative HPLC used a Shimadzu Prominence HPLC System with SPD-20A/20AV Series Prominence HPLC UV-Vis Detectors (Shimadzu, Tokyo, Japan).

NMR. NMR spectra, including ¹H-¹H COSY, HSQC, HMBC, and ROESY experiments, were carried out using a Varian UNITY INOVA 800 NMR spectrometer operating at 800 MHz (¹H) and 200 MHz (¹³C), with chemical shifts given in ppm (δ). MestReNova ver. 12.0.1 was used to process the data for the NMR spectra.

Analytical characterization. Experimental ECD spectra in MeOH were acquired in a quartz cuvette of 1 mm optical path length on a JASCO J-1500 spectropolarimeter (Tokyo, Japan). IR spectra were acquired on a Bruker IFS-66/S FT-IR spectrometer. Optical rotations were obtained utilizing a Jasco P-1020 polarimeter (Jasco, Easton, MD, USA).

Isolation of xylasporins. *Pseudoxyllaria* sp. X802 was prepared by inoculating 50 mL PDB with 2 agar cubes of a 2-week old *Pseudoxyllaria* sp. X802 culture. The culture was incubated for 2 weeks at 30 °C and 150 rpm. Afterwards, the pre-culture was used to inoculate 120 PDA plates (20 mL PDA, 92x16 mm), which were kept at 26 °C for 14 days in the dark. Agar plates were cut into pieces, soaked in EtOAc overnight. The organic extract was filtered off and the agar pieces were subsequently soaked in MeOH overnight, and filtrated again. The organic extracts were combined and dried *under vacuo*. The fungus crude extract (9.5 g) was suspended in distilled water (700 mL) and successively solvent-partitioned with EtOAc (700 mL) three times, yielding 2.0 g of residue. The EtOAc-soluble fraction was loaded onto silica-gel column chromatography and fractionated using a gradient solvent system of CH₂Cl₂-MeOH (100:1-1:1, v/v) to give five fractions (A-E). The fraction B (479 mg) was fractionated by preparative reversed-phase HPLC (Phenomenex Luna C18, 250 x 21.2 mm i.d., 5 μ m) using CH₃CN-H₂O (1:9–1:0, v/v, gradient system, flow rate: 5 mL/min) to provide six subfractions (B1–B6). Subfraction B4 (56 mg) was isolated by a semi-preparative reversed-phase HPLC (Phenomenex Luna C18, 250 x 10.0 mm i.d., 5 μ m) with 70% MeOH/H₂O (isocratic system, flow rate: 2 mL/min) to yield

compound **7** (1.7 mg, $t_R = 47.0$ min). The fraction C (160 mg) was separated by preparative reversed-phase HPLC (Phenomenex Luna C18, 250 × 21.2 mm i.d., 5 μ m) using CH₃CN-H₂O (1:9–1:0, v/v, gradient system, flow rate: 5 mL/min) to yield five subfractions (C1–C5). Compound **1** (1.4 mg, $t_R = 24.0$ min) was isolated from subfraction C3 (40 mg) by a semi-preparative reversed-phase HPLC (Phenomenex Luna C18, 250 × 10.0 mm i.d., 5 μ m) with 40% MeOH/H₂O (isocratic system, flow rate: 2 mL/min). Fraction C4 (20 mg) was purified by a semi-preparative reversed-phase HPLC (Phenomenex Luna C18, 250 × 10.0 mm i.d., 5 μ m) with 45% MeOH/H₂O (isocratic system, flow rate: 2 mL/min) to afford compound **2** (2.2 mg, $t_R = 30.0$ min). The fraction D (101 mg) was subjected onto by preparative reversed-phase HPLC (Phenomenex Luna C18, 250 × 21.2 mm i.d., 5 μ m) and separated by CH₃CN-H₂O (2:8–1:0, v/v, gradient system, flow rate: 5 mL/min) to yield five subfractions (D1–D5). Subfraction D2 (18 mg) was isolated by a semi-preparative reversed-phase HPLC (Phenomenex Luna C18, 250 × 10.0 mm i.d., 5 μ m) utilizing 60% MeOH/H₂O (isocratic system, flow rate: 2 mL/min) to give compounds **4** (1.2 mg, $t_R = 31.0$ min), **3** (1.5 mg, $t_R = 39.0$ min), **6** (1.8 mg, $t_R = 45.0$ min), and **5** (2.7 mg, $t_R = 48.0$ min).

Physical chemical data

Xylasporin A (**1**): Pale yellow oil; $[\alpha]^{25,D} 15.1$ (c 0.09, MeOH); IR (KBr) ν_{\max} 3379, 2921, 1786, 1652, 1048 cm⁻¹; ECD (MeOH) λ_{\max} ($\Delta\epsilon$) 228 (1.7), 265 (4.1) nm; ¹H (800 MHz) and ¹³C NMR (200 MHz), see Tables S1 and S2, respectively; positive HR-ESI-MS m/z 709.3430 [2M+H]⁺ (Calcd. for C₃₆H₅₃O₁₄, 709.3435).

Xylasporin B (**2**): Pale yellow oil; $[\alpha]^{25,D} -7.4$ (c 0.05, MeOH); IR (KBr) ν_{\max} 3422, 2919, 1784, 1552, 1248 cm⁻¹; ECD (MeOH) λ_{\max} ($\Delta\epsilon$) 245 (3.7) nm; ¹H (800 MHz) and ¹³C NMR (200 MHz), see Tables S1 and S2, respectively; positive HR-ESI-MS m/z 293.1742 [M-H₂O+H]⁺ (Calcd. for C₁₇H₂₅O₄, 293.1753).

Xylasporin C (**3**): Pale yellow oil; $[\alpha]^{25,D} -11.2$ (c 0.08, MeOH); IR (KBr) ν_{\max} 3412, 2903, 1779, 1498, 1228 cm⁻¹; ECD (MeOH) λ_{\max} ($\Delta\epsilon$) 242 (3.3) nm; ¹H (800 MHz) and ¹³C NMR (200 MHz), see Tables S1 and S2, respectively; positive HR-ESI-MS m/z 621.3627 [2M+H]⁺ (Calcd. for C₃₄H₅₃O₁₀, 621.3639).

Xylasporin D (**4**): Pale yellow oil; $[\alpha]^{25,D} 8.7$ (c 0.06, MeOH); IR (KBr) ν_{\max} 3405, 2895, 1770, 1454, 1213 cm⁻¹; ECD (MeOH) λ_{\max} ($\Delta\epsilon$) 251 (3.2) nm; ¹H (800 MHz) and ¹³C NMR (200 MHz), see Tables S1 and S2, respectively; positive HR-ESI-MS m/z 351.1784 [M+Na]⁺ (Calcd. for C₁₇H₂₈O₆Na, 351.1784).

Xylasporin E (**5**): Pale yellow oil; $[\alpha]^{25,D} 5.3$ (c 0.04, MeOH); IR (KBr) ν_{\max} 3412, 2903, 1787, 1464, 1223 cm⁻¹; ECD (MeOH) λ_{\max} ($\Delta\epsilon$) 245 (3.6) nm; ¹H (800 MHz) and ¹³C NMR (200 MHz), see Tables S1 and S2, respectively; positive HR-ESI-MS m/z 657.3850 [2M+H]⁺ (Calcd. for C₃₄H₅₇O₁₂, 657.3850).

Xylasporin F (**6**): Pale yellow oil; $[\alpha]^{25,D} 8.1$ (c 0.05, MeOH); IR (KBr) ν_{\max} 3402, 2895, 1780, 1458, 1215 cm⁻¹; ECD (MeOH) λ_{\max} ($\Delta\epsilon$) 245 (3.4) nm; ¹H (800 MHz) and ¹³C NMR (200 MHz), see Tables S1 and S2, respectively; positive HR-ESI-MS m/z 657.3855 [2M+H]⁺ (Calcd. for C₃₄H₅₇O₁₂, 657.3850).

Conformational search and geometry optimization. All conformers were acquired through Marvin module (ChemAxon). Minimize molecular conformations were obtained using an industry standard MMFF94 force field and a generic Dreiding force field. Conformers proposed in this study within 5 kJ/mol found in the MMFF force field were selected for geometry optimization by Gaussian 16 program package at B3LYP/6-31+G(d,p) level. The solvent (MeOH) effect was considered in the PCM approximation.

Chemical shifts calculation and DP4+ analysis. Geometrically optimized conformers for possible diastereomers of **1-6** were proceeded to calculation of gauge-invariant atomic orbital (GIAO) magnetic shielding tensors at the B3-LYP/6-31+G(d,p) level using Gaussian 16 program package.³⁶ Chemical shift values were calculated from the magnetic shielding tensors using **equation 1** where δ is the calculated NMR chemical shift for nucleus x , and σ^o is the shielding tensor for the proton and carbon nuclei in tetramethylsilane calculated with the B3-LYP/6-31+G(d,p) basis set.

Equation 1:
$$\delta_{calc}^x = \sigma^o - \sigma^x$$

The calculated unscaled NMR properties of the optimized structures were averaged based on the Boltzmann populations of conformers and scaled chemical shifts values were obtained using **equation 2**.

Equation 2:
$$\delta_{scaled} = \frac{\delta_{unscaled} - intercept}{slope}$$

Each experimental set of data was separately compared in detail with calculated ones, specifically for each accounted atom. Experimental and calculated chemical shifts (δ) were compared using the $\Delta\delta$ parameter according **equation 3**.

Equation 3:
$$\Delta\delta = |\delta_{calc} - \delta_{exp}|$$

where $\bar{\delta}_{calc}$ and $\bar{\delta}_{exp}$ are the calculated and experimental chemical shift values, respectively.

After calculating all $\Delta\delta$ values, mean absolute error (MAE) values and DP4+ probabilities were computed for determining which calculated set of data fits better with the experimental one. The MAE is defined as the summation (Σ) of the n computed absolute δ error values ($\Delta\delta$) normalized to the number of $\Delta\delta$ errors considered (n) (**equation 4**).

Equation 4:
$$\text{MAE} = \frac{\Sigma(\Delta\delta)}{n}$$

DP4+ probability analysis was processed upon using an Excel sheet from Grimblat *et al.*¹⁵

ECD spectra calculation. ECD calculations of **1** conformers were performed at the identical theory level and basis sets with Gaussian 16 program package.³⁰ Calculated ECD spectra were simulated by overlying each transition, where σ is the width of the band at the height $1/e$ (**equation 5**). E_i and R_i are the excitation energies and rotatory strengths for transition i , respectively. In the present study, the value of σ was 0.30 eV. The excitation energies and rotational strengths for ECD spectra were calculated, and ECD visualization was performed using GaussView 6.0.

Equation 5:
$$\Delta\varepsilon(E) = \sum_{i=1}^n \varepsilon_i(E) = \sum_{i=1}^n \left(\frac{R_i E_i}{2.29 \times 10^{-39} \sqrt{\pi} \sigma} \exp \left[- \left(\frac{E - E_i}{\sigma} \right)^2 \right] \right)$$

All DFT and TDDFT calculations were performed using Azzurra High-Performance Computing from Université Côte d'Azur and OPAL infrastructure.

Acknowledgements

This work was supported by the National Research Foundation of Korea (NRF) grant funded by the Korea government (MSIT) (2019R1A5A2027340 and 2021R1A2C2007937) and a New Faculty Research Grant of Pusan National University, 2023. This work was also supported under the framework of international cooperation program managed by the National Research Foundation of Korea (2020K2A9A2A06037042). This study was funded by the German Research Foundation (DFG, Deutsche Forschungsgemeinschaft) under Project-ID 239748522 – CRC 1127 (project A6), project BE 4799/4-1 under Project-ID 438841444, and the Germany's Excellence Strategy under Project-ID 390713860 - EXC 2051. We thank Mrs. Heike Heinecke (Hans Knöll Institute) for recording NMR spectra and Mrs. Andrea Perner (Hans Knöll Institute) for HRMS measurements. We thank Dr. Fabien Fontaine-Vive for launching DFT and TDDFT calculations as well as Université Côte d'Azur and OPAL infrastructure for providing Azzurra High-Performance Computing.

Keywords: polyketides • terpenes • fungal natural products • biosynthesis • metabolomics

Author contribution

SRL, MD, JF, HG, FS, JSY, CB designed research;

SRL, MD, JF, HG, FS, JSY, performed the experiments;

SRL, MD, JF, HG, FS, JSY, SYJ, CB, KHK analysed data;

SRL, MD, JF, HG, FS, JSY, CB generated the figures and

SRL, MD, JF, KHK, CB wrote the manuscript with input from all authors

Data availability

Supplementary Information contains additional details to experimental and analytical data.

References

- ¹ Spatafora JW, Aime MC, Grigoriev IV, Martin F, Stajich JE, Blackwell M. The Fungal Tree of Life: from Molecular Systematics to Genome-Scale Phylogenies. *Microbiol Spectr*. 2017;5(5).
- ² Becker K, Stadler M. Recent progress in biodiversity research on the Xylariales and their secondary metabolism. *J. Antibiot*. (Tokyo). 2021;74(1):1-23.
- ³ Rokas A, Mead ME, Steenwyk JL, Raja HA, Oberlies NH. Biosynthetic gene clusters and the evolution of fungal chemodiversity. *Nat. Prod. Rep*. 2020;37(7):868-878.
- ⁴ Kuhnert E, Navarro-Muñoz JC, Becker K, Stadler M, Collemare J, Cox RJ. Secondary metabolite biosynthetic diversity in the fungal family *Hypoxyloaceae* and *Xylaria hypoxylon*. *Stud. Mycol*. 2021;99:100118.
- ⁵ Schmidt S, Kildgaard S, Guo H, Beemelmans C, Poulsen M. The chemical ecology of the fungus-farming termite symbiosis. *Nat. Prod. Rep*. 2022;39:231–248.
- ⁶ Li H, Young SE, Poulsen M, Currie CR. **Symbiont-Mediated Digestion of Plant Biomass in Fungus-Farming Insects**. *Annu Rev. Entomol*. 2021;66:297–316.
- ⁷ Visser AA, Ros VI, De Beer ZW, Debets AJ, Hartog E, Kuyper TW, Laessøe T, Slippers B, Aanen DK. Levels of specificity of *Xylaria* species associated with fungus-growing termites: a phylogenetic approach. *Mol. Ecol*. 2009;18(3):553-67.
- ⁸ Visser AA, Kooij PW, Debets AJM, Kuyper TW, Aanen DK. **Pseudoxyllaria** as stowaway of the fungus-growing termite nest: Interaction asymmetry between *Pseudoxyllaria*, *Termitomyces* and free-living relatives. *Fung. Ecol*. 2021;4:322–332.
- ⁹ Wisselink M, Aanen DK, van 't Padje A. **The Longevity of Colonies of Fungus-Growing Termites and the Stability of the Symbiosis**. *Insects* 2020;11:527.
- ¹⁰ Fricke J, Schalk F, Kreuzenbeck NB, Seibel E, Hoffmann J, Dittmann G, Conlon BH, Guo H, ~~Wilhelm de Beer Z~~, Vassão DG, Gleixner G, Poulsen M, Beemelmans C. Adaptations of **Pseudoxyllaria** towards a comb-associated lifestyle in fungus-farming termite colonies. *ISME J*. 2023;17(5):733-747.
- ¹¹ Guo H, Kreuzenbeck NB, Otani S, Garcia-Altare M, Dahse HM, Weigel C, Aanen DK, Hertweck C, Poulsen M, Beemelmans C. **Pseudoxyllallemycins A-F, Cyclic Tetrapeptides with Rare Allenyl Modifications Isolated from Pseudoxyllaria sp. X802: A Competitor of Fungus-Growing Termite Cultivars**. *Org Lett*. 2016;18(14):3338-3341.
- ¹² Schalk F, Fricke J, Um S, Conlon BH, Maus H, Jäger N, Heinzl T, Schirmeister T, Poulsen M, Beemelmans C. GNPS-guided discovery of xylacremolide C and D, evaluation of their putative biosynthetic origin and bioactivity studies of xylacremolide A and B. *RSC Adv*. 2021;11(31):18748-18756.

- ¹³ Schalk F, Um S, Guo H, Kreuzenbeck NB, Görls H, de Beer ZW, Beemelmans C. Targeted Discovery of Tetrapeptides and Cyclic Polyketide-Peptide Hybrids from a Fungal Antagonist of Farming Termites. *Chembiochem*. 2020;21(20):2991–2996.
- ¹⁴ Adnani N, Rajski SR, Bugni TS. Symbiosis-inspired approaches to antibiotic discovery. *Nat. Prod. Rep.* 2017;34(7):784–814.
- ¹⁵ Keller, N. P. Fungal secondary metabolism: regulation, function and drug discovery. *Nat. Rev. Microbiol.* 2019 17, 167–180.
- ¹⁶ Wang, M. *et al.* Sharing and community curation of mass spectrometry data with Global Natural Products Social Molecular Networking. *Nat. Biotechnol.* 2016, 34, 828–837.
- ¹⁷ Gehrt A, Erkel G, Anke T, Sterner O. Cycloepoxydon, 1-hydroxy-2-hydroxymethyl-3-pent-1-enylbenzene and 1-hydroxy-2-hydroxymethyl-3-pent-1,3-dienylbenzene, new inhibitors of eukaryotic signal transduction. *J. Antibiot. (Tokyo)* 1998, 51, 455–463.
- ¹⁸ Ciavatta ML, Lopez-Gresa MP, Gavagnin M, Nicoletti R, Manzo E, Mollo E, Guo YW, Cimino G. Cytosporin-related compounds from the marine-derived fungus *Eutypella scoparia*. *Tetrahedron* 2008; 64:5365–5369.
- ¹⁹ Grimblat N, Zanardi MM, Sarotti AM. Beyond DP4: an Improved Probability for the Stereochemical Assignment of Isomeric Compounds using Quantum Chemical Calculations of NMR Shifts. *J. Org. Chem.* 2015; 80:12526–12534.
- ²⁰ Stevens-Miles S, Goetz MA, Bills GF, Giacobbe RA, Tkacz JS, Chang RS, Mojena M, Martin I, Diez MT, Pelaez F, Hensens OD, Jones T, Burg RW, Kong YL, Huang L. Discovery of an angiotensin II binding inhibitor from a *Cytospora* sp. using semi-automated screening procedures. *J. Antibiot. (Tokyo)*. 1996;49(2):119-123.
- ²¹ Liu Z, Jensen PR, Fenical W. A cyclic carbonate and related polyketides from a marine-derived fungus of the genus *Phoma*. *Phytochemistry* 2003;64:571–574.
- ²² Mehta G, Roy S. Enantioselective Total Synthesis of (+)-Eupenoxide and (+)-Phomoxide: Revision of Structures and Assignment of Absolute Configuration. *Org. Lett.* 2004;6:2389–2392.
- ²³ Akone, S. H., El Amrani, M., Lin, W., Lai, D. & Proksch, P. Cytosporins F–K, new epoxyquinols from the endophytic fungus *Pestalotiopsis theae*. *Tetrahedron Lett.* 2013; 54:6751–6754.
- ²⁴ Rukachaisirikul, V. *et al.* γ -Butyrolactone, cytochalasin, cyclic carbonate, eutypinic acid, and phenalenone derivatives from the soil fungus *Aspergillus* sp. PSU-RSPG185. *J. Nat. Prod.* 2014;77:2375–2382 (2014).
- ²⁵ Liao HX, Sun DW, Zheng CJ, Wang CY. A new hexahydrobenzopyran derivative from the gorgonian-derived Fungus *Eutypella* sp. *Nat. Prod. Res.* 2017;31:1640–1646.
- ²⁶ Zhang YX, Yu HB, Xu WH, Hu B, Guild A, Zhang JP, Lu XL, Liu XY, Jiao BH. Eutypellacytosporins A-D, Meroterpenoids from the Arctic Fungus *Eutypella* sp. D-1. *J. Nat. Prod.* 2019;82(11):3089-3095.
- ²⁷ Rivera-Chávez J, Zacatenco-Abarca J, Morales-Jiménez J, Martínez-Aviña B, Hernández-Ortega S, Aguilar-Ramírez E. Cuautepestorin, a 7,8-Dihydrochromene-Oxoisochromane Adduct Bearing a Hexacyclic Scaffold from *Pestalotiopsis* sp. IQ-011. *Org Lett.* 2019;21(10):3558-3562.
- ²⁸ Yu X, Müller WEG, Meier D, Kalscheuer R, Guo Z, Zou K, Umeokoli BO, Liu Z, Proksch P. Polyketide Derivatives from Mangrove Derived Endophytic Fungus *Pseudopestalotiopsis theae*. *Mar. Drugs.* 2020;18(2):129.
- ²⁹ Zhang YH, Du HF, Gao WB, Li W, Cao F, Wang CY. Anti-inflammatory Polyketides from the Marine-Derived Fungus *Eutypella scoparia*. *Mar. Drugs* 2022;20(8):486.
- ³⁰ Davis RA, Andjic V, Kotiw M, Shivas RG. Phomoxins B and C: Polyketides from an endophytic fungus of the genus *Eupenicillium*. *Phytochemistry* 2005; 66:2771–2775.
- ³¹ Nies J, Ran H, Wohlgemuth V, Yin WB, Li SM. Biosynthesis of the Prenylated Salicylaldehyde Flavoglucin Requires Temporary Reduction to Salicyl Alcohol for Decoration before Reoxidation to the Final Product. *Org. Lett.* 2020;22:2256–2260.
- ³² Terlouw BR, Blin K, Navarro-Muñoz JC, Avalon NE, Chevrette MG, Egbert S, Lee S, Meijer D, Recchia MJ, Reitz ZL, van Santen JA, Selem-Mojica N, Tørring T, Zaroubi L, Alanjary M, Aleti G, Aguilar C, Al-Salihi SAA, Augustijn HE, Avelar-

Rivas JA, Avitia-Domínguez LA, Barona-Gómez F, Bernaldo-Agüero J, Bielinski VA, Biermann F, Booth TJ, Carrion Bravo VJ, Castelo-Branco R, Chagas FO, Cruz-Morales P, Du C, Duncan KR, Gavriilidou A, Gayraud D, Gutiérrez-García K, Haslinger K, Helfrich EJN, van der Hoof JJJ, Jati AP, Kalkreuter E, Kalyvas N, Kang KB, Kautsar S, Kim W, Kunjapur AM, Li YX, Lin GM, Loureiro C, Louwen JJR, Louwen NLL, Lund G, Parra J, Philmus B, Pourmohsenin B, Pronk LJJ, Rego A, Rex DAB, Robinson S, Rosas-Becerra LR, Roxborough ET, Schorn MA, Scobie DJ, Singh KS, Sokolova N, Tang X, Udway D, Vigneshwari A, Vind K, Vromans SPJM, Waschulin V, Williams SE, Winter JM, Witte TE, Xie H, Yang D, Yu J, Zdouc M, Zhong Z, Collemare J, Linington RG, Weber T, Medema MH. MIBiG 3.0: a community-driven effort to annotate experimentally validated biosynthetic gene clusters. *Nucleic Acids Res.* 2023;51(D1):D603-D610.

³³ Liu L, Tang MC, Tang Y. Fungal Highly Reducing Polyketide Synthases Biosynthesize Salicylaldehydes That Are Precursors to Epoxycyclohexenol Natural Products. *J. Am. Chem. Soc.* 2019;141:19538–19541.

³⁴ Zhao Z, Ying Y, Hung YS, Tang Y. Genome Mining Reveals *Neurospora crassa* Can Produce the Salicylaldehyde Sordarial. *J. Nat. Prod.* 2019;82:1029–1033.

³⁵ Shannon, P. *et al.* Cytoscape: a software environment for integrated models of biomolecular interaction networks. *Genome Res* 2003;13:2498–2504.

³⁶ Frisch, M. J. *et al.* Gaussian 16 Rev. C.01. (2016).

Point-to-point for manuscript: Molecular networking analysis and computational NMR analysis uncovers six polyketide-terpene hybrids of the stowaway fungus *Pseudoxyllaria*

Reviewer #1	
This study reports a series of putative terpene-polyketide hybrids from a fungus of the genus Xylaria, which is here referred to under the subgenus name "Pseudoxyllaria". The editors and authors should be advised that strictly speaking, this is not correct according to the standing rules of taxonomy! If a genus is subdivided into subgenera and sections, the species placed in these subgeneric taxa retain the genus name.	We appreciate your comment and regret any deviation from strict adherence to fungal taxonomy regarding the fungus Pseudoxyllaria. The term "Pseudoxyllaria" has been commonly used in the literature due to the extensive history of biological research on termite-associated fungal strains. However, the precise taxonomic classification remains unresolved. Although our previous study in ISMJE 2023 highlighted genomic differences, similarities in phylogenomic markers and morphological characteristics suggest the likelihood of a subgenus existence. For clarity and to prevent further confusion concerning the strains, we have maintained the designation "subgeneric taxon Pseudoxyllaria" within this study. As stated in the introduction, we reference it based on its association with termite habitats. In case the editorial team advises us to use the genus name "Xylaria", rather than the more historical and ecology-driven name "Pseudoxyllaria", we are happy to change the text accordingly.
I also tried to find further information on the "strain X802" in reference 7 but failed to do so. That paper (Visser et al 2009) does not apparently give any information as to where the strains have been deposited and a strain 802 is not listed. Ref. 7 should actually not have been published in such a renowned journal, since it normally regarded absolutely important to deposit vouchers as a prerequisite for publication there. If the strain is indeed derived from the old study that was published ca. 15 years ago, the authors should clarify the origin and also explicitly state where this fungus is deposited. It could well be that it is extant in the FABI collection as it was originally obtained from South Africa and scientists from FABI are co-authors of the respective publications, or as the first author is Dutch it could have ended up with CBS/WFBI?	We apologize for the lack of clarity from the outset, which likely arose from our oversight in not including references 10 and 11 in the same position as the paper by Visser et al. (2009). While the original paper by Visser et al. (2009) does not specifically designate the fungus as X802, the first ITS sequence of these strains was reported in that study. Subsequently, the X802 strains have been utilized as a model in various publications, which includes collaborations with the University of Pretoria (de Beer), Wageningen University (Visser, Aanen), and the University of Copenhagen (Poulsen). Please note that we also share several publications with the senior author of the publication "Visser et al 2009". Therefore, we have made it clear in the reference list of this and other articles that this strain has consistently been a part of

	collaborative research efforts and the FABI collection. We would also point out our references 7-13 in this manuscript draft for further details on previous work.
This said, I am already at the end of my criticism because the way in which the new compounds were isolated, identified and characterized is state of the art. The structure elucidation seems very sound, perhaps except for the ECD calculations to somewhat tentatively determine the absolute configuration, which can never replace a total synthesis or an X-ray structure. There are already some examples showing that this ECD calc. stuff does not always work out. On the other hand, it seems that these calculations have become a kind of new fashion and other reviewers may actually insist that such data be included. There are even IR data included, which shows that the authors were very careful to provide as much information on their new compounds as possible. The MS-MS based dereplication in combination with 2D-MMR data and DP4+ probability analyses is a rather interesting and innovative approach.	We greatly value the constructive feedback provided. We acknowledge that relying solely on ECD calculations may not be the optimal approach for determining absolute configuration, particularly in the absence of intense chromophores. This is why we chose to conduct ECD calculations for only one compound, where most stereocenters were determined by NMR calculations, leaving only a few unassigned. Therefore, for compound 1, determining the absolute stereochemistry necessitates a comprehensive approach involving 2D NMR, NMR calculations, comparison, as well as ECD measurements and calculations.
The theory about the biosynthetic origin deduced from transcriptomics and genome analyses is also very sound, even though purists may say that this methodology can ultimately not replace classical feeding experiments. The cited literature is almost complete and very comprehensive regarding the cytosporins and other related metabolites. The discussion and the conclusion are very sound. The whole ms is rather well written aside from some typos and the spectral data in the SI seem to support the structure proposals very well.	We appreciate the comments and acknowledge the value of stable isotope-feeding as an insightful approach to deduce certain biosynthetic steps, an approach we have extensively documented in the literature. However, when dealing with PKS-terpenoid hybrids, it is highly probable that cross-feeding occurs between different (primary) metabolic pathways. As a result, for example, ¹³C labeling may not yield a comprehensive read-out.
The whole ms is rather well written aside from some typos and the spectral data in the SI seem to support the structure proposals very well.	We greatly acknowledge the critical reading of the manuscript and SI.
I have made some small annotations in the pdf file and recommend the paper for acceptance after minor revisions.	We have included the comments into the revision. Very much appreciated.
Reviewer #2	
Although the novelty of the structures and biosynthesis of the compounds is somewhat lacking in impact, the methods employed and the analytical data are reliable from a chemical point of view, and there appears to be a certain value in publishing them as a paper. However, the following points should be improved or revised.	We highly appreciate the thorough review of both the manuscript and the supplementary information.

1. In Abstract. I am not sure about the last sentence in the abstract. First of all, I don't think the expression is grammatically very favorable. Second, is the analysis/discussion you have done related to the biosynthetic origin of xylasporins?	We regret any inaccuracies in our wording and have amended the statement to better convey the intended message.
2. In Figure 1. As can be said throughout this paper, there is no correspondence between each image and the text, such as A-E in the main figure. On the other hand, there is a detailed correspondence with SI. Figure 1 is somewhat better, but the subsequent figures are completely unable to guess the correspondence with the text. These points require significant improvement. Going back to Figure 1, I have no idea why Fig. 1B and Fig. 1C need to appear here. I can guess that Fig1E is related to the culture method, but please mention it in more detail in the text.	We apologize for the lack of clear referencing of the Figures in the text. We have improved the referencing of the Figures throughout the manuscript and ensured that each Subfigure is mentioned in the text at its first occurrence. For Figure 1 specifically, we changed the order of figures to improve the readability of the text. All details have been stated in the figure legend as well.
3. In Figure 3. The carbon No. in the structural formula written in Fig. 3C is off. Please correct the main text to correspond to Fig. 3A-C. I am not sure what the dashed squares mean. Please map it to the main text. 4. In Figure 4. Please add that the arrows indicate NOESY correlations in the figure of the relative three-dimensional arrangement in Fig. 4A. Please map the main text properly to Fig 4. 5. In Figure 5. Please respond in the same way as Fig. 4.	Many thanks for pointing this out. We have revised the figures and checked the cross-referencing across the manuscript.
6. In Figure 6. The possible biosynthetic pathway is proposed in the second half of Fig. 6, explained in paragraph P8, but it is too brief and it is difficult to understand how each step is done. Please try to explain in more detail	We have elaborated on the putative pathway more in details and hope that these efforts help the understanding of the section.
Reviewer #3 (Remarks to the Author):	
Overall, this is very solid work, in particular concerning the structure elucidation. The paper is well written and illustrated. However, the identified compounds are structurally highly related to the long-known cytosporins, essentially only diverging in the length of the C8 side-chain (being shorter in the new compound series). No new biosynthetic insights or exciting biological activities are reported. The degree of novelty of the findings might thus be considered relatively now. I still believe that the findings are interesting to natural product discovery groups, particularly also to chemical biologists. Acceptance of the work in Communications Chemistry still seems borderline in its current state.	We appreciate the positive and critical constructive feedback of the reviewer. Indeed the structural scaffolds resemble those of the cytosporins. However, the structural features of the cytosporins have been up to discussion with several yet non-clarified structural assignments especially with respect to the absolute stereochemistry. In this article, we have comprehensively summarized these features and also consolidated the knowledge of known compounds for comparative analysis. First, this study discusses all three subgroups of

[...] Currently, publication in Sci. Rep. or a specialized natural product chemistry journal (JNP) would seem more appropriate.	metabolites can be depicted (epoxy variants, trihydroxy derivatives and those carrying a carbamate moiety), all of which have been found in this study. Secondly, this study made the effort to calculate NMR data of more than 50 stereoisomers of these three categories, which goes far beyond the studies of more specialized journals, which usually just report a subset of the data presented here. In addition, the measured and calculated NMR datasets will help and guide future studies related to this compound family. Thirdly, although the existence of fog-like clusters has been previously reported, the assembly line depicted herein enables the correlation of the herein isolated derivatives. The overall considerations will facilitate the future targeted design of molecular biological studies focused on either individual enzyme classes or the entire pathway. Fourthly, In frame of the revision of this manuscript, we were now able to elaborate on the structural features of xylasporin I (8), whose planar structure has been reported earlier. However, due to its instability, we were unable to report the relative stereochemistry yet. Using a more targeted approach, we were now able to provide sufficient NMR-based evidence to allow the deduction of its relative stereochemistry also with support of GIAO NMR chemical shifts calculations of more than 50 stereoisomers. Lastly, and while serendipitously, we were able to determine and revise the relative and absolute structure of the co-secreted xylacremolide B (9) by single X-ray analysis. Thus, we consider the overall dataset of importance for the broader readership.
If the authors could provide more biological data (broader testing against fungi – also with relevance to the natural environment of the producer strain; further profiling against bacteria / cytotoxicity evaluation), the significance of the work could be considerably improved. [...] In addition, the subchapter “antibacterial activity studies” seems to only contain a single antifungal test system -> please adjust.	This suggestion holds significant value, and we are keen to explore additional ecological assays. However, at present, Pseudoxylaria strains do not produce spores in laboratory cultures, making germination assays unfeasible. As a result, we have opted to utilize Aspergillus, which naturally appears when the comb/colony is weakened, as a suitable model fungus to investigate the effects of metabolites on germination.

	Furthermore, we acknowledge that the activity assays against bacteria that yielded negative results were not included in the experimental section. We apologize for this oversight and have subsequently revised the bioactivity section in the manuscript and the Material/Methods part to include these negative findings. The weak antimicrobial inhibitory activity of only few of the isolated secondary metabolites supported furthermore previous conclusion that Pseudoxylaria might have adapted to such comb-specific stressors by reducing and specializing the secondary metabolome to minimize triggers that could stimulate alarm responses of the fungal mutualist and termites.
Minor comments: there are some minor typos / missing words throughout the manuscript.	Thank you for the observation. We have revised and proof read the complete manuscript

REVIEWERS' COMMENTS:

Reviewer #1 (Remarks to the Author):

It is absolutely irrelevant if the fungus has been wrongly named in previous papers that were not reviewed by capable persons.

Repeating previous errors does not make them go away.

The genus name *Pseudoxylaria* is not valid and must not be used.

The most comprehensive phylogenetic study available as yet is that by Hsieh et al. (2010) who used the name in the sense of a subgenus.

It is about time to correct this bad mistake!

All other comments including those by the other reviewers seem to have been answered in a satisfactory manner.

Reviewer #2 (Remarks to the Author):

I judged that the areas I pointed out were handled with care. However, I have one additional point to make.

In Fig7B. The movement of the electrons in compound V is unclear. Note exactly where they start and where they go.

Reviewer #3 (Remarks to the Author):

I appreciate the explanations and additional experimental insights provided by the authors in response to the requests raised by the reviewers. The paper should be accepted for publication.

Second point-to-point response: Molecular networking analysis and computational NMR analysis uncovers six polyketide-terpene hybrids from termite-associated *Xylaria* isolates.

Reviewer #1 (Remarks to the Author):	
It is absolutely irrelevant if the fungus has been wrongly named in previous papers that were not reviewed by capable persons. Repeting previous errors does not make them go away. The genus name Pseudoxylaria is not valid and must not be used. The most comprehensive phylogenetic study available as yet is that by Hsieh et al. (2010) who used the name in the sense of a subgenus. It is about time to correct this bad mistake!	As suggested, we have now changed Pseudoxylaria to Xylaria and do understand the argument of the reviewer and the taxonomic implications. Thus, we have also changed the title “Molecular networking analysis and computational NMR analysis uncovers six polyketide-terpene hybrids from termite-associated Xylaria isolates” to clarify that members of the subgenus are addressed. It also needs to be noted that the change of the name was also done within the Supporting Information.
All other comments including those by the other reviewers seem to have been answered in a satisfactory mmanner.	We thank the reviewer for the time to evaluate the manuscript
Reviewer #2 (Remarks to the Author):	
I judged that the areas I pointed out were handled with care. However, I have one additional point to make. In Fig7B. The movement of the electrons in compound V is unclear. Note exactly where they start and where they go.	We apologize for the unclear depiction and have corrected the arrows related to oxidation state and electron shifts.
Reviewer #3 (Remarks to the Author):	
I appreciate the explanations and additional experimental insights provided by the authors in response to the requests raised by the reviewers. The paper should be accepted for publication.	We thank the reviewer for the time to evaluate our manuscript